# Impact of different frequencies of controlled breath and pressure-support levels during biphasic positive airway pressure ventilation on the lung and diaphragm in experimental mild acute respiratory distress syndrome

Alessandra F. Thompson[1,2], Lillian Moraes[1], Nazareth N. Rocha[1,3], Marcos V. S. Fernandes[1], Mariana A. Antunes[1], Soraia C. Abreu[1], Cintia L. Santos[1], Vera L. Capelozzi[4], Cynthia S. Samary[1,5], Marcelo G. de Abreu[6,7], Felipe Saddy[1,2,8], Paolo Pelosi[9,10], Pedro L. Silva[1‡], Patricia R. M. Rocco[1‡]*

1 Laboratory of Pulmonary Investigation, Carlos Chagas Filho Biophysics Institute, Federal University of Rio de Janeiro, Rio de Janeiro, RJ, Brazil, 2 Copa D'Or Hospital, Rio de Janeiro, Brazil, 3 Department of Physiology and Pharmacology, Biomedical Institute, Fluminense Federal University, Niterói, Brazil, 4 Department of Pathology, School of Medicine, University of São Paulo, São Paulo, Brazil, 5 Department of Physical Therapy, Federal University of Rio de Janeiro, Rio de Janeiro, RJ, Brazil, 6 Department of Anesthesiology and Intensive Care Therapy, Pulmonary Engineering Group, University Hospital Dresden, Technische Universität Dresden, Dresden, Germany, 7 Outcomes Research Consortium, Cleveland, OH, United States of America, 8 Pró-Cardíaco Hospital, Rio de Janeiro, Brazil, 9 Department of Surgical Sciences and Integrated Diagnostics, University of Genoa, Genoa, Italy, 10 San Martino Policlinico Hospital, IRCCS for Oncology and Neurosciences, Genoa, Italy

‡ PLS and PRMR share senior authorship on this work
* prmrocco@biof.ufrj.br

**Data Availability Statement:** All relevant data are within the paper and its Supporting Information

## Abstract

### Background

We hypothesized that a decrease in frequency of controlled breaths during biphasic positive airway pressure (BIVENT), associated with an increase in spontaneous breaths, whether pressure support (PSV)-assisted or not, would mitigate lung and diaphragm damage in mild experimental acute respiratory distress syndrome (ARDS).

### Materials and methods

Wistar rats received *Escherichia coli* lipopolysaccharide intratracheally. After 24 hours, animals were randomly assigned to: 1) BIVENT-100+PSV$_{0\%}$: airway pressure (P$_{high}$) adjusted to V$_T$ = 6 mL/kg and frequency of controlled breaths (*f*) = 100 bpm; 2) BIVENT-50+PSV$_{0\%}$: P$_{high}$ adjusted to V$_T$ = 6 mL/kg and *f* = 50 bpm; 3) BIVENT-50+PSV$_{50\%}$ (PSV set to half the P$_{high}$ reference value, i.e., PSV$_{50\%}$); or 4) BIVENT-50+PSV$_{100\%}$ (PSV equal to P$_{high}$ reference value, i.e., PSV$_{100\%}$). Positive end-expiratory pressure (P$_{low}$) was equal to 5 cmH$_2$O. Nonventilated animals were used for lung and diaphragm histology and molecular biology analysis.

files. The MATLAB files and LabVIEW diagram block are available at https://github.com/PedroLolo81/Matlab_Routines_for_Lung.git.

**Funding:** This study was supported by Conselho Nacional de Desenvolvimento Científico e Tecnológico in the form of grants awarded to PRMR (421067/2016-0) and PLS (443894/2018-3), and Fundação Carlos Chagas Filho de Amparo à Pesquisa do Estado do Rio de Janeiro in the form of grants awarded to PRMR (E-26/210.910/2016, E-26/010.001488/2019).

**Competing interests:** The authors have declared that no competing interests exist.

## Results

BIVENT-50+$PSV_{0\%}$, compared to BIVENT-100+$PSV_{0\%}$, reduced the diffuse alveolar damage (DAD) score, the expression of amphiregulin (marker of alveolar stretch) and muscle atrophy F-box (marker of diaphragm atrophy). In BIVENT-50 groups, the increase in PSV (BIVENT-50+$PSV_{50\%}$ *versus* BIVENT-50+$PSV_{100\%}$) yielded better lung mechanics and less alveolar collapse, interstitial edema, cumulative DAD score, as well as gene expressions associated with lung inflammation, epithelial and endothelial cell damage in lung tissue, and muscle ring finger protein 1 (marker of muscle proteolysis) in diaphragm. Transpulmonary peak pressure (Ppeak,L) and pressure–time product per minute ($PTP_{min}$) at $P_{high}$ were associated with lung damage, while increased spontaneous breathing at $P_{low}$ did not promote lung injury.

## Conclusion

In the ARDS model used herein, during BIVENT, the level of PSV and the phase of the respiratory cycle in which the inspiratory effort occurs affected lung and diaphragm damage. Partitioning of inspiratory effort and transpulmonary pressure in spontaneous breaths at $P_{low}$ and $P_{high}$ is required to minimize VILI.

## Introduction

Inappropriate mechanical ventilation settings in patients with the acute respiratory distress syndrome (ARDS) may result in ventilation-induced lung injury (VILI). VILI is believed to involve a proinflammatory response, leading to lung structural and peripheral organ damage [1]. The use of protective low tidal volume under controlled mechanical ventilation is the only ventilator strategy known to reduce mortality in ARDS [2]. However, controlled mechanical ventilation may lead to diaphragmatic weakness [3,4], thus delaying the weaning process [3]. Partial ventilatory support can be implemented in mild-to-moderate forms of ARDS [5–7]. Since it requires less sedation and no neuromuscular blockade, it prevents muscle atrophy [8] and is associated with better cardiovascular performance [9,10], shorter time on mechanical ventilation, and shorter intensive care unit (ICU) stay [10,11]. On the other hand, spontaneous breathing during assisted mechanical ventilation may aggravate lung injury, since it can increase patient-ventilator asynchrony and work of breathing, leading to so-called patient self-inflicted lung injury (P-SILI) [12–14]. In recent decades, different partial ventilatory support modes have been proposed [15]. During biphasic positive airway pressure ventilation (BIVENT), a combination of time-cycled controlled breaths at two levels of continuous positive airway pressure and spontaneous breathing is allowed at both low and high airway pressure phases [16]. In experimental ARDS, Saddy et al. reported reduced lung and diaphragm damage with lower frequency of controlled breaths during BIVENT [17]. The combination of BIVENT with pressure support ventilation (PSV), when compared with pressure controlled ventilation, has been found to reduce lung damage [18]. We hypothesized that a decrease in frequency of control breaths during BIVENT, associated with an increase in spontaneous breaths, whether pressure support (PSV)-assisted or not, would mitigate lung and diaphragm damage in ARDS. The present study evaluated respiratory variables, histology, biological markers associated with VILI, and markers of diaphragmatic injury under different frequencies of control breaths and PSV in a rat model of experimental mild ARDS.

## Materials and methods

### Study approval

This study was approved by the Ethics Committee of the Healthy Science Center (CEUA no. 103/16), Federal University of Rio de Janeiro, Rio de Janeiro, Brazil. All animals received humane care in compliance with the "Principles of Laboratory Animal Care" formulated by the National Society for Medical Research and the Guide for the Care and Use of Laboratory Animals prepared by the U.S. National Academy of Sciences. The present study followed the ARRIVE guidelines for reporting of animal research [19]. Animals came from the Breeding Facility of Healthy Science Center of Federal University of Rio de Janeiro. Conventional animals were housed at a controlled temperature (23˚C) and controlled light–dark cycle (12–12 h), with free access to water and food. No acclimation was done.

### Animal preparation and experimental protocol

Forty male Wistar rats (mean weight 292±20g) were anesthetized under spontaneous breathing with 1.5–2.0% isoflurane (Isoforine®; Cristália, Itapira, SP, Brazil) and subjected to intratracheal instillation of $9.6\times10^6$ EU/mL *Escherichia coli* lipopolysaccharide (Merck Millipore, Burlington, Massachusetts, USA), diluted in 200 μL of 0.9% saline solution.

After 24 h, animals were premedicated intraperitoneally (i.p.) with midazolam (1–2 mg/kg) and anesthetized with ketamine (100 mg/kg, i.p.). An intravenous (i.v.) catheter (Jelco 24G, Becton, Dickinson and Company, New Jersey, NJ, USA) was inserted into the tail vein, and anesthesia induced and maintained with midazolam (2 mg/kg/h) and ketamine (50 mg/kg/h). During spontaneous breathing, anesthetic depth was evaluated by the response to light touch with a fingertip on the rat's whiskers (0 = awake, fully responsive to surroundings; 1 = not responsive to surroundings, rapid response to whisker stimulation; 2 = slow response; 3 = unresponsive to whisker stimulation) [20], pupil diameter, position of the nictitating membrane, and movement in response to tail stimulation [21,22]. Experiments were started when responses to a noise stimulus (handclap), whisker stimulation, and tail clamping were absent. The depth of the anesthesia was monitored via mean arterial pressure, heart rate and respiratory rate throughout the experiment.

Body temperature was maintained at 37.5 ± 1˚C with a heating bed (EFF 421, INSIGHT®, Brazil). After local infiltration of 0.4 mL lidocaine (1%), a tracheostomy was performed and a polyethylene cannula (PE 240, Intramedic®, Clay-Adams Inc, New York, USA; internal diameter 1.8 mm, length 7.5 cm) was introduced into the trachea. A second catheter (18G; Arrow International, USA) was then placed in the right internal carotid artery for blood sampling and gas analysis (Radiometer ABL80 FLEX, Copenhagen NV, Denmark), as well as monitoring of mean arterial pressure (MAP) (Networked Multiparameter Veterinary Monitor LifeWindow 6000 V; Digicare Animal Health, Boynton Beach, FL, USA). Animals were adapted to an airway pressure transducer (UT-PDP-70; SCIREQ, Canada) and a two-sidearm pneumotachograph (internal diameter 2.7 mm, length 25.7 mm, internal volume 0.147 ml, airflow resistance 0.0057 cm $H_2O\cdot ml^{-1}\cdot s^{-1}$) [23] connected to a differential pressure transducer (UT-PDP-02, SCIREQ, Montreal, QC, Canada), for airflow (V') measurement. A 30-cm-long water-filled catheter (PE-205; Becton, Dickinson and Company) with side holes at the tip, connected to a differential pressure transducer (UT-PL-400; SCIREQ, Canada), was used to measure the esophageal pressure. Briefly, the esophageal catheter was passed into the stomach and then slowly returned into the esophagus; its proper positioning was assessed using the "occlusion test" [24].

Animals were mechanically ventilated (SERVO-i; MAQUET, Solna, Sweden) in assisted pressure-controlled ventilation (A-PCV) with ΔP set to achieve a tidal volume ($V_T$) of 6 mL/

kg, positive end-expiratory pressure (PEEP) of 0 $cmH_2O$, I:E (inspiratory: expiratory ratio) of 1:2, respiratory rate (RR) of 100 breaths per minute (bpm), and $FiO_2$ (inspired oxygen fraction) of 0.4 at BASELINE-ZEEP, to evaluate whether the degree of lung damage was similar between ARDS groups. Flow trigger sensitivity was adjusted at BASELINE-PEEP (INITIAL) for adequate inspiratory effort, according to esophageal pressure variation (ΔPes). No additional changes to flow trigger sensitivity were made at any point during the experiment [25]. Shortly thereafter (defined as the INITIAL time point), animals were randomly assigned to one of four groups of BIVENT:

1. BIVENT-100+$PSV_{0\%}$ (n = 8) with $P_{high}$ to achieve $V_T$ = 6 mL/kg, time at high and low pressures ($T_{high}$, and $T_{low}$, respectively) of 0.3 s, RR 100 bpm;

2. BIVENT-50+$PSV_{0\%}$ (n = 8), with $P_{high}$ to achieve $V_T$ = 6 mL/kg, $T_{high}$ and $T_{low}$ of 0.3 and 0.9 s, respectively, RR 50 bpm;

3. BIVENT-50+$PSV_{50\%}$ (n = 8) with $P_{high}$ to achieve $V_T$ = 6 mL/kg, pressure support ventilation of half the $P_{high}$ value ($PSV_{50\%}$), $T_{high}$ and $T_{low}$ of 0.3 and 0.9 s, respectively, RR 50 bpm;

4. BIVENT-50+$PSV_{100\%}$ (n = 8), with $P_{high}$ to achieve $V_T$ = 6 mL/kg, pressure support ventilation equal to $P_{high}$ ($PSV_{100\%}$), $T_{high}$ and $T_{low}$ of 0.3 and 0.9 s, respectively, RR 50 bpm (**Fig 1A**).

$P_{high}$ was adjusted across all groups to achieve $V_T$ = 6 mL/kg, while PSV adjustments were 50% or 100% of the $P_{high}$ level adjusted for each animal. Spontaneous breathing activity was allowed during all ventilatory strategies, including BIVENT-100+$PSV_{0\%}$. Sedation and anesthesia were adjusted to keep adequacy of inspiratory efforts during mechanical ventilation. The $P_{low}$ level, which reflects PEEP, was set at 5 $cmH_2O$, based on previous observations from our group showing that higher PEEP levels would lead to deterioration in respiratory mechanics in a similar rat model of ARDS [26]. We did not discriminate whether PSV occurred at $P_{low}$ vs. $P_{high}$, since the SERVO-i ventilator enables PSV only during $T_{low}$. In all groups, $FiO_2$ = 0.4 was maintained for 1 hour, at which time blood gas analysis (Radiometer, Copenhagen, Denmark) and mechanical data were obtained (timepoint FINAL) (**Fig 1B**). At timepoint FINAL, heparin was injected (1,000 IU i.v.), and animals were euthanized by overdose of sodium thiopental (60 mg/kg i.v.; Cristália, Brazil). The trachea was clamped at $P_{low}$ = 5 $cmH_2O$, lungs were removed *en bloc* for histology and molecular biology analysis, and a surgical line was placed in the left bronchus to maintain lung volume at $P_{low}$ = 5 $cmH_2O$. The right lung was immediately frozen in liquid nitrogen for molecular biology analyses. The diaphragm was also removed at the end of the experiments. Eight of 40 rats were instilled with *E. coli* LPS, but not ventilated (NV); these animals were used for molecular biology analysis.

## Data acquisition and respiratory system mechanics

Airflow, airway pressure (Paw), and Pes were recorded continuously throughout the experiments by a computer running custom-made software written in LabVIEW (National Instruments, USA). All signals were amplified in a three-channel signal conditioner (TAM-DHSE Plugsys Transducers Amplifiers, Module Type 705/2, Harvard Apparatus, Holliston, Massachusetts, USA) and sampled at 200 Hz with a 12-bit analog-to-digital converter (National Instruments; Austin, Texas, USA). All mechanical data were computed offline by a routine written in MATLAB (Version R2007a; The Mathworks Inc., USA) (Please see in the

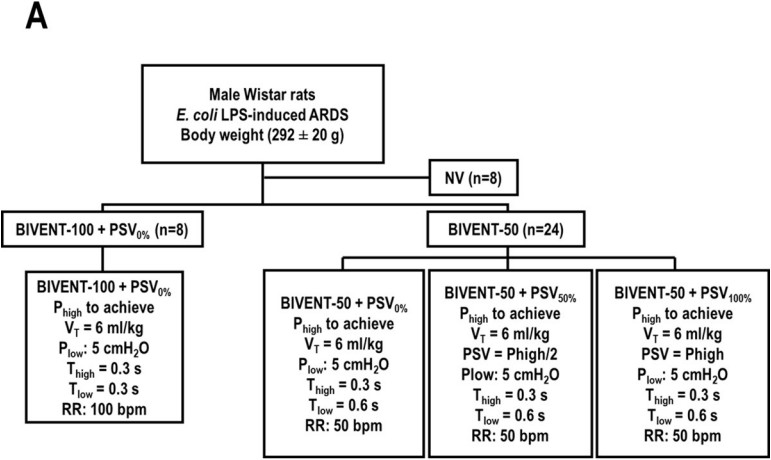

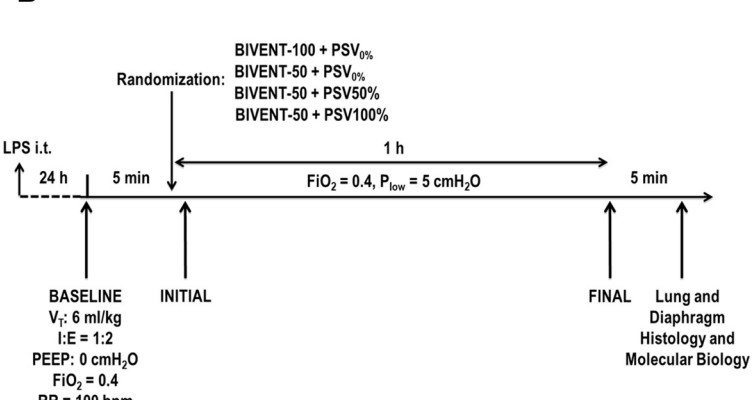

**Fig 1. A.** Experimental design. BIVENT-100+PSV0% (n = 8) with Phigh to achieve VT = 6 mL/kg, Time at high and low pressures ($T_{high}$ and $T_{low}$, respectively) = 0.3 s, RR = 100 bpm; BIVENT-50+PSV0% (n = 8), with $P_{high}$ to achieve $V_T$ = 6 mL/kg, Thigh and Tlow = 0.3 and = 0.9 s, respectively, RR = 50 bpm; BIVENT-50+PSV$_{50\%}$ (n = 8) with Phigh to achieve $V_T$ = 6 mL/kg, pressure support ventilation of half value of $P_{high}$ (PSV$_{50\%}$), Thigh and Tlow = 0.3 and 0.9 s, respectively, RR = 50 bpm; and BIVENT-50+PSV100% (n = 8) with Phigh to achieve $V_T$ = 6 mL/kg, pressure support ventilation equal $P_{high}$ (PSV$_{100\%}$), $T_{high}$ and $T_{low}$ = 0.3 and = 0.9 s, respectively, RR = 50 bpm. ARDS: Acute respiratory distress syndrome; BIVENT: Biphasic positive airway pressure; NV: Nonventilated; LPS: Lipopolysaccharide; $P_{high}$: High level of continuous positive airway pressure; $P_{low}$: Low level of continuous positive airway pressure. $T_{high}$: Time spent in $P_{high}$; $T_{low}$: Time spent in $P_{low}$; RR: Respiratory rate; PSV: Pressure support ventilation. B. Timeline of the experiments. i.t.: Intratracheal; $V_T$: Tidal volume; I:E: Inspiratory-to-expiratory ratio; PEEP: Positive end-expiratory pressure; FiO$_2$: Fraction of inspired oxygen.

supplement custom-made software written in LabVIEW and routine written in MATLAB for data analysis).

$V_T$ was calculated by digital integration of the flow signal. Coefficient of variation (CV) of $V_T$ was determined among 600 sampled cycles by the ratio of standard deviation divided by mean values of $V_T$. The total respiratory rate (RR) was calculated from the Pes swings as the frequency per minute of each type of breathing cycle. Mean transpulmonary pressure (Pmean, L) and peak transpulmonary pressure (Ppeak,L) were calculated as the difference between tracheal and esophageal pressure. Inspiratory time divided by total respiratory cycle time (Ti/Ttot) was calculated. The pressure–time product per minute (PTP$_{min}$) was calculated as the

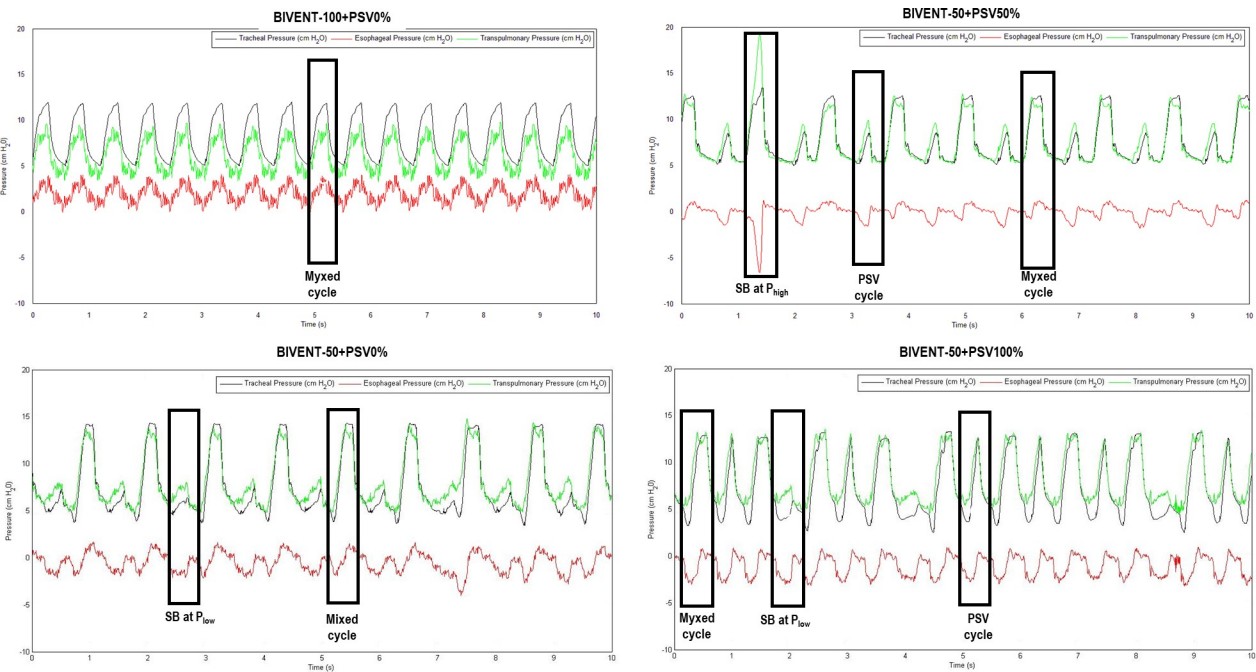

**Fig 2. Original tracheal, esophageal and transpulmonary pressure tracings.**

integral of ΔPes over one minute. The ΔPes reflects the total variation of esophageal pressure during the inspiratory effort. All mechanical parameters were extracted from four different types of breathing cycles as follows: 1) mixed respiratory cycles (M), i.e., negative Pes swings with simultaneous ventilator inspiratory cycling; 2) spontaneous breath cycles, without PSV, at high airway pressure ($P_{high}$), i.e., negative Pes swings at $P_{high}$ not followed by ventilator cycling; 3) spontaneous breath cycles, without PSV, at low airway pressure ($P_{low}$), i.e., negative Pes swings at $P_{low}$ not followed by ventilator cycling; and 4) spontaneous breath at pressure support (PSV), only present in groups BIVENT-50+PSV$_{50\%}$ and BIVENT-50+PSV$_{100\%}$ during the $T_{low}$ phase (**Fig 2**).

## Histology

**Diffuse alveolar damage.** The left lung was fixed in 4% formaldehyde solution and embedded in paraffin. Sections (4 μm thick) were cut longitudinally from the central zone with a microtome and stained with hematoxylin–eosin for histologic analysis. Photomicrographs at magnifications of ×25, ×100, and ×400 were obtained from eight non-overlapping fields of view per section under a light microscope (Olympus BX51; Olympus Latin America Inc., Brazil). Diffuse alveolar damage (DAD) score was quantified by an expert in lung pathology (V.L.C.) blinded to group assignment [27]. Briefly, scores of 0 to 4 were used to represent overdistension, interstitial edema, and alveolar collapse, with 0 standing for no effect and 4 for maximum severity. Additionally, the extent of each scored characteristic per field of view was determined on a scale of 0 to 4, with 0 standing for no visible evidence and 4 for complete involvement. Scores were calculated as the product of severity and extent of each feature, on a range of 0 to 16. The cumulative DAD score was the sum of these three features and thus ranged from 0 to 48 [28].

**Electron microscopy.** Three slices measuring 2×2×2 mm were cut from three different segments of the right lung and from the right diaphragm and fixed in 2.5% glutaraldehyde and

0.1 M phosphate buffer (pH = 7.4) for transmission electron microscopy (TEM) (JEOL 1010 Transmission Electron Microscope, Tokyo, Japan). Each TEM image (20 per animal) was analyzed for damage to epithelial and endothelial cells, basement membrane, and extracellular matrix at three different magnifications. Pathologic findings were graded on a 5-point semiquantitative severity-based scoring system as follows: 0 = normal lung parenchyma, 1 = changes in 1% to 25% of examined tissue, 2 = changes in 26% to 50% of examined tissue, 3 = changes in 51% to 75% of examined tissue, and 4 = changes in 76% to 100% of examined tissue [29]. For diaphragm analysis, the following aspects were assessed on TEM: 1) myofibril abnormalities, defined as disruption of myofibril bundles or disorganized myofibrillar pattern with edema of the Z-disc and 2) mitochondrial injury with abnormal, swollen mitochondria and abnormal cristae. The pathologic findings were again graded on a 5-point semiquantitative severity-based scoring system, as follows: 0 = normal diaphragm, 1 = changes in 1% to 25%, 2 = changes in 26% to 50%, 3 = changes in 51% to 75%, and 4 = changes in 76% to 100% of examined tissue. The pathologist working on light microscopy and TEM images (V.L.C.) was blinded to group assignment.

**Molecular biology analysis of lung and diaphragm tissue.** Quantitative real-time reverse transcription polymerase chain reaction was performed to measure biological markers associated with inflammation (tumor necrosis factor [TNF]-α), alveolar stretch (amphiregulin), epithelial cell damage (club cell protein 16), endothelial cell damage [vascular cell adhesion molecule (VCAM)-1], and extracellular matrix damage (decorin) in lung tissue, as well as markers of muscle proteolysis [muscle RING finger-1 (MuRF-1) and muscle atrophy F-box (MAFbx/atrogin-1)] in the right diaphragm. The primer sequences are listed in **S1 Table**. Central slices of right lung and right diaphragm were cut, collected in cryotubes, flash-frozen by immersion in liquid nitrogen, and stored at −80°C. Total RNA was extracted from frozen tissues using the RNeasy Plus Mini Kit (Qiagen, Hilden, Germany), following the manufacturer's recommendations. The RNA concentration was measured by spectrophotometry in a Nanodrop ND-2000 system. First-strand cDNA was synthesized from total RNA using a Quantitec reverse transcription kit (Qiagen, Hilden, Germany). Relative mRNA concentrations were measured with a SYBR green detection $_{system}$ using ABI 7500 real-time polymerase chain reaction (Applied Biosystems, Foster City, CA, USA). Samples were measured in triplicate. For each sample, the expression of each gene was normalized to that of the housekeeping gene *36B4* (acidic ribosomal phosphoprotein P0) and expressed as fold change relative to NV, using the $2^{-\Delta\Delta Ct}$ method, where $\Delta Ct$ = Ct (reference gene)–Ct (target gene). All analyses were performed by two authors (M.A.A., C.L.S.), who were blinded to group assignment.

## Statistical analysis

The sample size was judiciously calculated to minimize the use of animals. A sample of 8 animals per group would provide the appropriate power (1-β = 0.8) to identify significant (α = 0.05) differences in alveolar collapse between BIVENT-100+PSV$_{0\%}$ and BIVENT50+PSV$_{0\%}$ [17], taking into account an effect size d = 1.72, a two-sided test, and a sample size ratio of 1 (G*Power 3.1.9.2, University of Düsseldorf, Germany). The Kolmogorov–Smirnov test with Lilliefors' correction was used to assess normality of data, while the Levene median test was used to evaluate the homogeneity of variances. For comparisons between BIVENT-100 and BIVENT-50 groups, either Student's *t*-test or the Mann–Whitney *U* test was used as appropriate. For comparisons within BIVENT-50 groups, one-way ANOVA with Holm-Šídák's post-hoc test (<0.05) or the Kruskal-Wallis test followed by Dunn's test were used. All tests were performed in GraphPad Prism v8.4.0 (GraphPad Software, La Jolla, CA, USA). Significance was established at $P < 0.05$ (two-sided).

## Results

There were no missing data at any time point in the study. Two animals died due to hemodynamic compromise during the pilot phase. At BASELINE ZEEP, $PaO_2/FiO_2$ was lower than 300 mmHg in all groups (S2 Table). MAP was higher than 70 mmHg throughout the experiments. At timepoint FINAL, $PaO_2/FiO_2$, pHa, $PaCO_2$, and $HCO_3^-$ did not significantly differ between BIVENT-100+PSV$_{0\%}$ and BIVENT-50+PSV$_{0\%}$, nor among the BIVENT-50+PSV$_{0\%}$, BIVENT-50+PSV$_{50\%}$, and BIVENT-50+PSV$_{100\%}$ groups (S3 Table). The amount of fluid infused did not differ between groups (S3 Table). Adjusted P$_{high}$ and PSV levels are shown in S4 Table.

### BIVENT-100+PSV$_{0\%}$ vs BIVENT-50+PSV$_{0\%}$ group

The CV of V$_T$ was lower in BIVENT-100+PSV$_{0\%}$ than BIVENT-50+PSV$_{0\%}$ (Table 1). Among the mixed cycles, BIVENT-100+PSV$_{0\%}$ presented higher Ti/Ttot and RR compared to BIVENT-50+PSV$_{0\%}$. No significant changes were observed in PTP$_{min}$ and ΔPes between groups.

NV animals showed higher alveolar collapse and cumulative DAD score compared to BIVENT-100+PSV$_{0\%}$ and BIVENT-50+PSV$_{0\%}$. The score of overdistension and interstitial edema was higher in NV than BIVENT-50+PSV$_{0\%}$. BIVENT-100+PSV$_{0\%}$ cumulative DAD score compared to BIVENT-50+PSV$_{0\%}$ (Fig 3, Table 2).

Endothelial cell damage was greater in NV compared to than BIVENT-50+PSV$_{0\%}$ (Fig 4 and Table 3).

BIVENT-100 + PSV$_{0\%}$ group showed increased amphiregulin gene expression in comparison to BIVENT-50+PSV$_{0\%}$ (Fig 5).

Myofibril abnormality score was higher in BIVENT-100+PSV$_{0\%}$ than NV and BIVENT-50 +PSV$_{0\%}$, mitochondrial injury did not differ among NV, BIVENT-100+PSV$_{0\%}$ and BIVENT-100+PSV$_{50\%}$ (Fig 6 and Table 4).

MAFbx gene expression was higher in BIVENT-100+PSV$_{0\%}$ than BIVENT-50+PSV$_{0\%}$ (Fig 7).

### Comparisons among the BIVENT-50 + PSV$_{0\%}$, BIVENT-50 + PSV$_{50\%}$, and BIVENT-50 + PSV$_{100\%}$ groups

Among total cycles, the CV of V$_T$ was lower in BIVENT-50+PSV$_{50\%}$ than BIVENT-50+PSV$_{0\%}$ and BIVENT-50+PSV$_{100\%}$ groups. In addition, Pmean,$_L$, PTP$_{min}$, and ΔPes were lower in BIVENT-50+PSV$_{100\%}$ than BIVENT-50 + PSV50% animals (Table 1). Among P$_{low}$ cycles, V$_T$, airflow, RR, and PTP$_{min}$ were lower in BIVENT-50+PSV100% than BIVENT-50+PSV$_{0\%}$. Among P$_{high}$ cycles, RR, Ppeak,$_L$, and PTP$_{min}$ were higher in BIVENT-50+PSV$_{50\%}$ than BIVENT-50+PSV$_{0\%}$. Among mixed (M) cycles, Ti/Ttot was higher, while Pmean,$_L$ was lower in BIVENT-50+ PSV$_{100\%}$ compared to BIVENT-50+PSV$_{50\%}$. Among PSV cycles, ΔPes was lower in BIVENT-50+PSV$_{100\%}$ than BIVENT-50+PSV$_{50\%}$ animals. Overdistension, alveolar collapse, and cumulative DAD score were higher in BIVENT-50+PSV$_{50\%}$ than BIVENT-50+-PSV$_{0\%}$, while BIVENT-50+PSV$_{100\%}$ showed less interstitial edema and alveolar collapse, as well as a lower cumulative DAD score, compared to BIVENT-50+PSV$_{50\%}$ (Fig 3, Table 2). BIVENT-50+PSV$_{50\%}$ showed more damage to epithelial and endothelial cells, basement membrane, and extracellular matrix compared to BIVENT-50 + PSV$_{0\%}$. The BIVENT-50 + PSV$_{100\%}$ group exhibited less basement membrane damage compared to BIVENT-50+-PSV$_{50\%}$ (Fig 4 and Table 3).

**Table 1. Respiratory mechanical parameters at timepoint FINAL.**

| Parameter | Cycle | BIVENT-100 + PSV$_{0\%}$ | BIVENT-50 | | |
|---|---|---|---|---|---|
| | | | BIVENT-50+PSV$_{0\%}$ | BIVENT-50+PSV$_{50\%}$ | BIVENT-50+PSV$_{100\%}$ |
| $V_T$ (mL/kg) | M | 5.8 ± 0.9 | 5.9 ± 1.0 | 6.2 ± 0.8 | 5.5 ± 0.9 |
| | PSV | - | - | 4.8 ± 1.0 | 5.0 ± 0.7 |
| | $P_{low}$ | 2.5 ± 1.4 | 3.9 ± 1.8 | - | 1.7 ± 0.6$^{\#}$ |
| | $P_{high}$ | - | 5.8 ± 0.9 | 7.2 ± 2.0 | - |
| | Total | 5.8 ± 1.3 | 5.2 ± 1.2 | 5.8 ± 0.7 | 5.0 ± 0.6 |
| CV of $V_T$ (%) | M | 14 ± 11 | 15 ± 13 | 12 ± 8 | 13 ± 3 |
| | PSV | - | - | 12 ± 5 | 16 ± 5 |
| | $P_{low}$ | 18 ± 20 | 23 ± 13 | - | 39 ± 18 |
| | $P_{high}$ | - | 51 ± 9 | 36 ± 19 | - |
| | Total | 4 ± 4 | 32 ± 11** | 12 ± 6# | 28 ± 9† |
| Airflow (mL/s) | M | 12.1 ± 3.6 | 11.9 ± 3.5 | 11.8 ± 1.9 | 11.5 ± 2.3 |
| | PSV | - | - | 8.5 ± 1.9 | 10.2 ± 1 |
| | $P_{low}$ | 6.4 ± 3.1 | 8.0 ± 2.7 | - | 3.7 ± 2.1$^{\#}$ |
| | $P_{high}$ | - | 10.2 ± 1.9 | 6.8 ± 1.8 | |
| | Total | 11.6 ± 1.8 | 10.5 ± 2.5 | 10.8 ± 1.7 | 10.2 ± 1.4 |
| RR (bpm) | M | 96 ± 4 | 51 ± 4** | 51 ± 2 | 49 ± 2 |
| | PSV | - | - | 48 ± 10 | 51 ± 9 |
| | $P_{low}$ | 18 ± 9 | 45 ± 16 | - | 9 ± 2$^{\#}$ |
| | $P_{high}$ | - | 5 ± 1 | 8 ± 1$^{\#}$ | - |
| | Total | 110 ± 6 | 93 ± 25 | 81 ± 28 | 113 ± 19 |
| Ppeak, L (cmH$_2$O) | M | 15.6 ± 3.8 | 13.7 ± 1.6 | 13.9 ± 2.9 | 14.6 ± 3.1 |
| | PSV | - | - | 12.1 ± 1.7 | 13.9 ± 3.2 |
| | $P_{low}$ | 11.8 ± 3.8 | 10.5 ± 2.3 | - | 10.6 ± 1.5 |
| | $P_{high}$ | - | 15.3 ± 1.2 | 22.1 ± 3.0# | - |
| | Total | 15.5 ± 3.4 | 12.3 ± 1.7 | 13.1 ± 2.1 | 13.8 ± 3.0 |
| Pmean, L (cmH$_2$O) | M | 5.9 ± 0.9 | 6.8 ± 1.4 | 7.4 ± 1.7 | 5.4 ± 0.6† |
| | PSV | - | - | 3.1 ± 0.5 | 3.4 ± 0.5 |
| | $P_{low}$ | 2.4 ± 1.3 | 3.5 ± 2.0 | - | 1.9 ± 0.6 |
| | $P_{high}$ | - | 2.8 ± 0.1 | 3.1 ± 0.4 | - |
| | Total | 5.6 ± 1.1 | 4.9 ± 1.3 | 6.1 ± 2.6 | 4.1 ± 0.3† |
| Ti/Ttot (s) | M | 0.6 ± 0.1 | 0.5 ± 0.1** | 0.4 ± 0.1 | 0.6 ± 0.1#† |
| | PSV | - | - | 0.5 ± 0.1 | 0.5 ± 0.1 |
| | $P_{low}$ | 0.6 ± 0.1 | 0.6 ± 0.1 | - | 0.6 ± 0.1 |
| | $P_{high}$ | - | 0.6 ± 0.1 | 0.6 ± 0.1 | - |
| | Total | 0.6 ± 0.1 | 0.5 ± 0.1 | 0.4 ± 0.1 | 0.6 ± 0.1† |
| PTP$_{min}$ (cmH$_2$O*sec/min) | M | 53 + 37 | 44 ± 20 | 42 ± 15 | 27 ± 16 |
| | PSV | - | - | 46 ± 16 | 35 ± 8 |
| | $P_{low}$ | 19 ± 19 | 43 ± 22 | - | 10 ± 2# |
| | $P_{high}$ | - | 2 ± 1 | 9 ± 4# | - |
| | Total | 67 ± 30 | 62 ± 26 | 50 ± 18 | 26 ± 18#† |

(*Continued*)

**Table 1.** (Continued)

| Parameter | Cycle | BIVENT-100 + PSV$_{0\%}$ | BIVENT-50 | | |
|---|---|---|---|---|---|
| | | | BIVENT-50+PSV$_{0\%}$ | BIVENT-50+PSV$_{50\%}$ | BIVENT-50+PSV$_{100\%}$ |
| ΔPes (cmH$_2$O) | **M** | 1.7 ± 1.4 | 1.4 ± 1.1 | 1.3 ± 1.2 | 1.7 ± 1.4 |
| | **PSV** | - | - | 3.3 ± 1.0 | 0.8 ± 0.5† |
| | **P$_{low}$** | 4.2 ± 2.9 | 3.7 ± 1.5 | - | 3.8 ± 1.5 |
| | **P$_{high}$** | - | 4.4 ± 0.3 | 7.3 ± 3.6 | - |
| | **Total** | 2.1 ± 1.5 | 1.9 ± 1.1 | 2.7 ± 0.9 | 1.2 ± 0.6† |

Values are given as mean ± standard deviation (SD) of 8 animals in each group. Comparisons between BIVENT-100+PSV$_{0\%}$ and BIVENT-50+PSV$_{0\%}$ groups were done using Student t-test ($p<0.05$).

**vs. BIVENT-100+PSV$_{0\%}$. Comparisons among BIVENT-50 groups were done using One-Way ANOVA followed by Holm-Šídák post hoc test ($p<0.05$)

# vs BIVENT-50 + PSV$_{0\%}$

† vs BIVENT-50+PSV50%.

BIVENT: Biphasic positive airway pressure at different rates of time-cycled controlled breaths: 100 and 50 breaths/min; PSV$_{0\%}$:no pressure support ventilation; PSV$_{50\%}$: Pressure support ventilation 50% P$_{high}$; PSV100%: Pressure support ventilation 100% P$_{high}$; M = mixed, assisted breaths; P$_{high}$ = spontaneous breaths at high continuous positive airway pressure; Plow: Spontaneous breaths at low continuous positive airway pressure; PSV: Pressure support ventilation; Total: Mean data for mixed, PSV, P$_{low}$, and P$_{high}$; V$_T$: Tidal volume; CV of V$_T$: Coefficient of variation of tidal volume; RR: Respiratory rate; Ppeak, L: Transpulmonary peak pressure; Pmean, L: Transpulmonary mean pressure; Ti/Ttot: Inspiratory time divided by total respiratory cycle time; PTP$_{min}$: Pressure–time product per minute; ΔPes: Esophageal pressure swing.

TNF-α, VCAM-1, amphiregulin, and decorin gene expressions were higher in BIVENT-50 + PSV$_{50\%}$ than BIVENT-50+PSV$_{0\%}$. On the other hand, BIVENT-50+PSV$_{100\%}$ showed reduced TNF-α, CC-16, and VCAM-1 gene expression compared to BIVENT-50+PSV$_{50\%}$

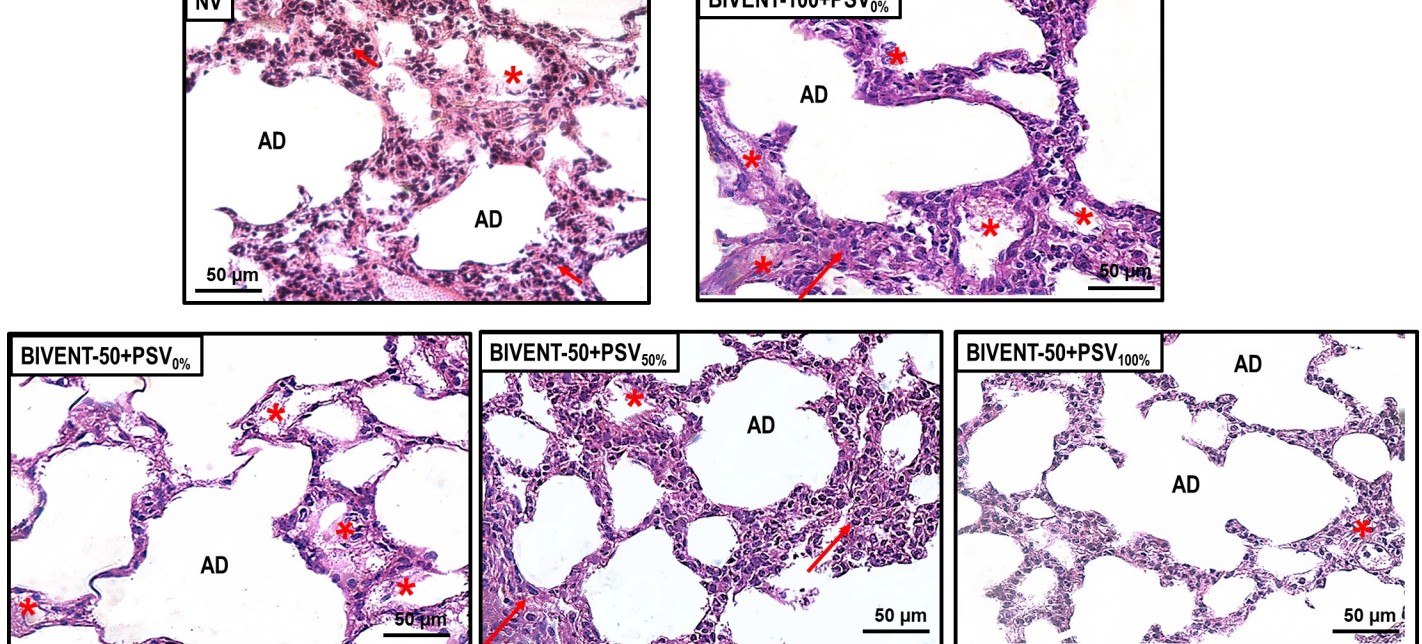

**Fig 3. Representative photomicrographs of lung parenchyma stained with hematoxylin–eosin.** NV (nonventilated), BIVENT-100+PSV$_{0\%}$ and BIVENT-50+PSV$_{0\%}$: The frequency of control breaths is 100 and 50 breaths/min, respectively. BIVENT-50+PSV$_{50\%}$: PSV set to half the value of P$_{high}$ (PSV$_{50\%}$). BIVENT-50+PSV$_{100\%}$: PSV equal to the value at P$_{high}$ (PSV$_{100\%}$). Note the preserved microscopic architecture of the lung parenchyma in BIVENT-50+PSV$_{50\%}$ animals. AD: Alveolar duct. Asterisk: Interstitial edema. Arrows: Areas of alveolar collapse. Scale bar = 50 μm.

**Table 2. Diffuse alveolar damage score.**

| Features of diffuse alveolar damage score | NV | BIVENT-100+PSV$_{0\%}$ | BIVENT-50 | | |
|---|---|---|---|---|---|
| | | | BIVENT-50+PSV$_{0\%}$ | BIVENT-50+PSV$_{50\%}$ | BIVENT-50+PSV$_{100\%}$ |
| Overdistension (0–16) | 4 (4–6) | 4 (2–4) | 2.5 (2–4)* | 6 (4.5–8.25)# | 5 (3.25–7.5) |
| Interstitial Edema (0–16) | 6 (6–8.75) | 4 (3–8.25) | 4 (3.25–4)* | 5 (4–6) | 2 (2–4)† |
| Alveolar Collapse (0–16) | 9 (6.5–11.25) | 5 (4–6)* | 3 (2–4)* ** | 6 (4–8.75)# | 2 (2–3.5)† |
| Cumulative DAD score (0–48) | 20 (19–22) | 12.5 (11–18.5)* | 9 (8.25–12)* ** | 18 (14.5–21)# | 10.5 (7.75–12)† |

Cumulative diffuse alveolar damage score (scores arithmetically averaged from two independent investigators) representing injury from variables: Overdistension, interstitial edema, and alveolar collapse. Values are given as median (interquartile range) of 8 animals in each group. Comparisons among NV, BIVENT-100+PSV$_{0\%}$, and BIVENT-50+PSV$_{0\%}$ groups as well as among BIVENT-50 groups were done by Kruskal-Wallis followed by Dunn's test. (p<0.05) *vs NV

**vs BIVENT-100+PSV$_{0\%}$

#vs BIVENT-50+PSV$_{0\%}$, †vs BIVENT-50+PSV50%. DAD: Diffuse alveolar damage. NV: Nonventilated. BIVENT: Biphasic positive airway pressure at different rates of time-cycled controlled breaths (100 and 50 breaths/min); PSV$_{0\%}$: No pressure support ventilation; PSV$_{50\%}$: Pressure support ventilation 50% P$_{high}$; PSV$_{100\%}$: Pressure support ventilation 100% P$_{high}$; P$_{high}$ = spontaneous breaths at high continuous positive airway pressure.

(Fig 5). The BIVENT-50+PSV$_{100\%}$ group exhibited higher amphiregulin expression than BIVENT-50+PSV$_{0\%}$ animals.

BIVENT-50+PSV$_{50\%}$ showed more myofibril abnormalities than BIVENT-50+PSV$_{0\%}$ and BIVENT-50+PSV$_{100\%}$ (Fig 6, Table 4). The mitochondrial injury score was higher in BIVENT-50+PSV$_{50\%}$ than BIVENT-50+PSV$_{0\%}$ and BIVENT-50+PSV$_{100\%}$. MURF-1 gene expression was higher in BIVENT-50+PSV$_{50\%}$ than in BIVENT-50+PSV$_{0\%}$ and BIVENT-50+-PSV$_{100\%}$ (Fig 7). No significant changes were observed in MAFbx expression among BIVENT-50 groups (Fig 7).

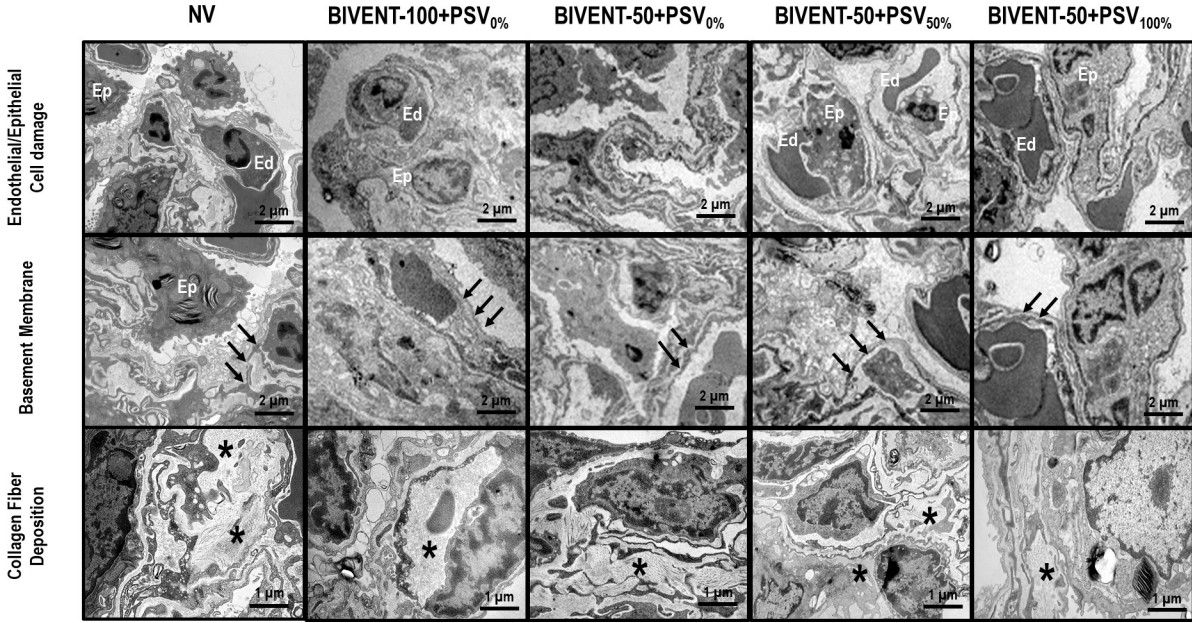

**Fig 4. Photomicrographs of electron microscopy of the lung.** The ultrastructure of the alveolar–capillary barrier shows varying degrees of injury to epithelial/endothelial cells and the basement membrane, as well as collagen fiber deposition in the septal interstitium. NV: Nonventilated. Note that BIVENT-50+PSV$_{50\%}$ induced more epithelial (Ep) and endothelial cell (Ed) apoptosis, irregularity and thickness of the basement membrane (arrows), and collagen fiber deposition (*) in the alveolar-capillary barrier than BIVENT-50+PSV$_{0\%}$. In contrast, less epithelial (Ep) and endothelial cell (Ed) apoptosis, greater basement membrane integrity (arrows), and less collagen fiber deposition (*) can be seen in BIVENT-50+PSV$_{100\%}$ compared to BIVENT-50+PSV$_{50\%}$.

**Table 3. Semiquantitative analysis of lung electron microscopy.**

| Features of lung electron microscopy | NV | BIVENT-100+PSV$_{0\%}$ | BIVENT-50 | | |
|---|---|---|---|---|---|
| | | | BIVENT-50+PSV$_{0\%}$ | BIVENT-50+PSV$_{50\%}$ | BIVENT-50+PSV$_{100\%}$ |
| **Endothelial cell damage** | 3 (2–3.25) | 2.5 (2–3) | 2 (1.75–2)* ** | 3 (2.75–3.25)# | 2.5 (2–3) |
| **Epithelial cell damage** | 3 (2–3.25) | 3.5 (2–4) | 2 (1.75–2.25)** | 3.5 (3–4)# | 3 (2.75–4) |
| **Basement membrane damage** | 3 (2–4) | 3 (2.75–3.25) | 2 (1.75–2.25)** | 3.5 (2.75–4)# | 2 (1.75–2.25)† |
| **ECM damage** | 2.5 (2–3) | 3 (2–3) | 2 (1–3) | 3.5 (3–4)# | 2.5 (1.75–3) |

Ultrastructure features of electron microscopy of the lung (scores arithmetically averaged from two independent investigators) representing injury from variables: Endothelial apoptosis, epithelial apoptosis, basement membrane damage and cumulative score. Values are given as median (interquartile range) of 8 animals in each group. Comparisons among NV, BIVENT-100+PSV$_{0\%}$, and BIVENT-50+PSV$_{0\%}$ groups as well as among BIVENT-50 groups were done by Kruskal-Wallis followed by Dunn's test. (p<0.05) *vs NV

**vs BIVENT-100+PSV$_{0\%}$.

#vs BIVENT-50+PSV$_{0\%}$

†vs BIVENT-50+PSV50%. ECM: Extracellular matrix. NV: Nonventilated. BIVENT: Biphasic positive airway pressure at different rates of time-cycled controlled breaths (100 and 50 breaths/min); PSV$_{0\%}$: No pressure support ventilation; PSV$_{50\%}$: Pressure support ventilation 50% P$_{high}$; PSV$_{100\%}$: Pressure support ventilation 100% P$_{high}$; P$_{high}$ = spontaneous breaths at high continuous positive airway pressure.

## Discussion

In the rat model of mild ARDS used herein, at a low protective V$_T$ (6 mL/kg), we found that the decrease in the frequency of controlled breaths (BIVENT-100+PSV$_{0\%}$ *versus* BIVENT-50

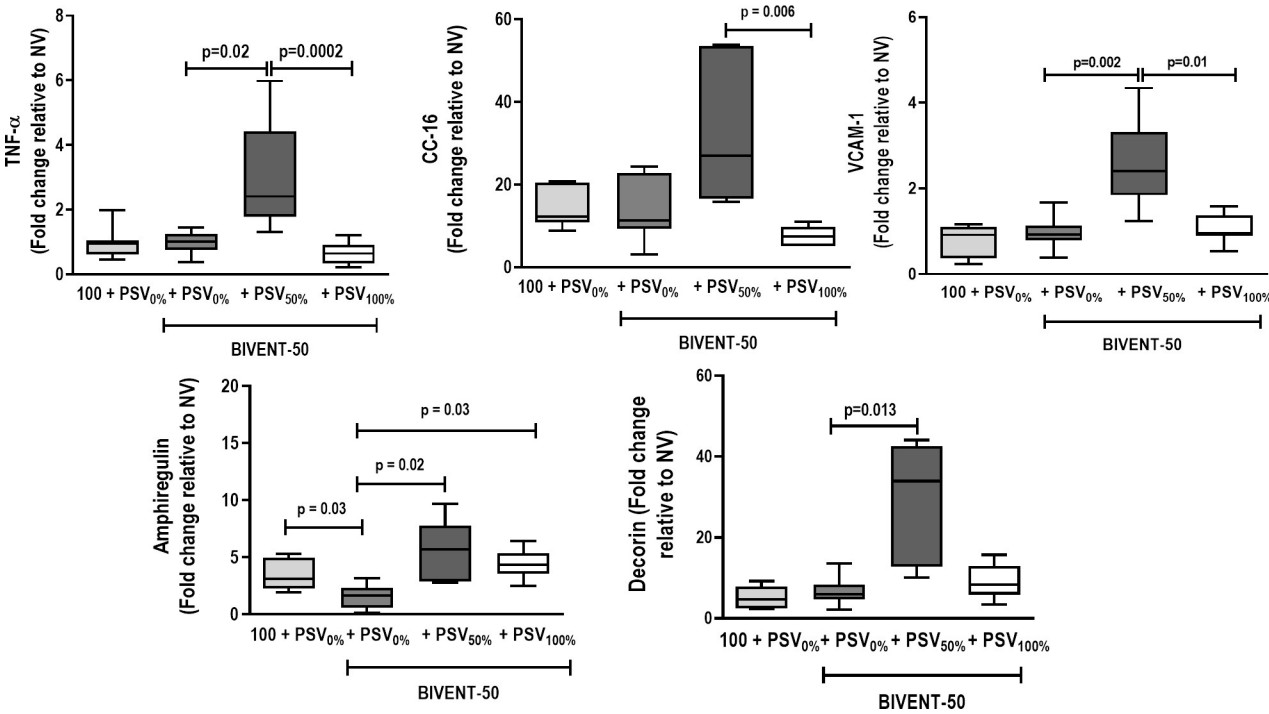

**Fig 5. Real-time polymerase chain reaction analysis of biological markers for inflammation (tumor necrosis factor [TNF]-α), epithelial cell damage (club cell secretory protein [CC-16]), endothelial cell damage (vascular cell adhesion molecule [VCAM]-1), alveolar stretch (amphiregulin), and extracellular matrix damage (decorin).** Box plots represent the median and interquartile range of 8 animals. Relative gene expression was calculated as a ratio of the average gene expression levels compared with the reference gene (*36B4*) and expressed as fold change relative to respective NV (nonventilated). Comparisons between BIVENT-100+PSV$_{0\%}$ and BIVENT-50+PSV$_{0\%}$ groups were done by the Mann–Whitney *U* test (p<0.05). For comparisons within BIVENT-50 groups, the Kruskal-Wallis test with Dunn's post-hoc test was used (p<0.05).

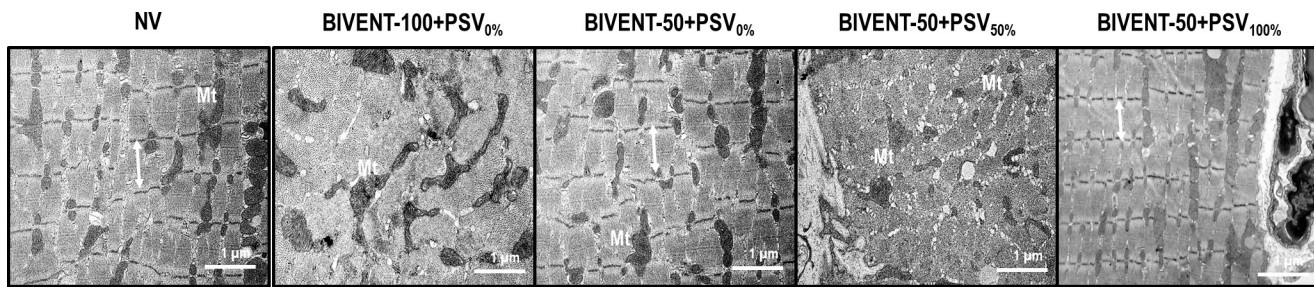

**Fig 6. Electron microscopy of the diaphragm.** Photomicrographs are representative of data obtained from diaphragm sections of eight animals per group. Myofibril damage with Z-disc edema and mitochondrial injury (Mt) was greater in BIVENT-100+PSV$_{0\%}$ compared to BIVENT-50+PSV$_{0\%}$. Diaphragmatic mitochondrial damage was more intense in BIVENT-50+PSV50% than BIVENT-50+PSV100%. Sarcomere disarrangement (double arrows) and Z-disc edema were more pronounced during BIVENT-50+PSV$_{50\%}$ compared to BIVENT-50+PSV$_{100\%}$. NV: Nonventilated.

+PSV$_{0\%}$) reduced DAD score, amphiregulin expression in lung tissue and MAFbx expression in diaphragm. In BIVENT-50 groups, the increase in PSV (BIVENT-50+PSV$_{50\%}$ *versus* BIVENT-50+PSV$_{100\%}$) yielded better lung mechanics and less alveolar collapse, interstitial edema, cumulative DAD score, basement membrane damage, as well as gene expressions of TNF-α, CC-16, and VCAM-1 in lung tissue, and MURF-1 expression in diaphragm. Transpulmonary peak pressure (Ppeak,L) and pressure–time product per minute (PTP$_{min}$), both at P$_{high}$, were associated with lung damage, while increased rate of spontaneous breaths at P$_{low}$ was not. In short, total values of PTP$_{min}$ (inspiratory effort) and Ppeak,L (transpulmonary pressure) did not contribute towards reduction of VILI; however, partitioning of these parameters between spontaneous breaths at P$_{low}$ and at P$_{high}$ is required during BIVENT to optimize ventilator settings.

We used a model of mild lung injury induced by intratracheal instillation of *E. coli* lipopolysaccharide (*E. coli* LPS) because it reproduces several characteristics of mild human ARDS [30]. We observed mean PaO$_2$/FiO$_2$ < 300 mmHg at BASELINE-ZEEP; nevertheless, in small animals, changes in lung function and histology (alveolar collapse, neutrophil infiltration, and edema) are more closely related to the degree of lung damage than oxygenation levels are [30]. The model used herein is a two-hit model: endotoxin (first hit) induced alveolar and interstitial edema, alveolar–capillary barrier changes, and elevated markers of inflammation within the first hour after tracheal instillation, increasing progressively until the 24-h timepoint, when

**Table 4. Semiquantitative analysis of diaphragm electron microscopy.**

| Features of diaphragm electron microscopy | NV | BIVENT-100+PSV$_{0\%}$ | BIVENT-50 | | |
|---|---|---|---|---|---|
| | | | BIVENT-50+PSV$_{0\%}$ | BIVENT-50+PSV$_{50\%}$ | BIVENT-50+PSV$_{100\%}$ |
| **Myofibril abnormality** | 2 (1–2) | 2.5 (2–3)* | 2 (1.25–2)** | 3 (2–3)# | 1.5 (1–2)† |
| **Mitochondrial injury** | 2 (2–2) | 2.5 (2–3) | 2 (2–2) | 3 (3–4)# | 2 (1–2)† |

Ultrastructure features of electron microscopy of the diaphragm (scores arithmetically averaged from two independent investigators) representing injury from these two variables: (1) myofibril abnormalities, defined as disruption of myofibril bundles or disorganized myofibrillar pattern with Z-disk edema, and (2) mitochondrial injury with abnormal swollen mitochondria and abnormal cristae. Values are given as median (interquartile range) of 8 animals in each group. Comparisons among NV, BIVENT-100+PSV$_{0\%}$, and BIVENT-50+PSV$_{0\%}$ groups as well as among BIVENT-50 groups were done by Kruskal-Wallis followed by Dunn's test. (p<0.05) *vs NV
**vs BIVENT-100+PSV$_{0\%}$.
#vs BIVENT-50+PSV$_{0\%}$
†vs BIVENT-50+PSV50%. NV: Nonventilated. BIVENT: Biphasic positive airway pressure at different rates of time-cycled controlled breaths (100 and 50 breaths/min); PSV$_{0\%}$: No pressure support ventilation; PSV$_{50\%}$: Pressure support ventilation 50% P$_{high}$; PSV$_{100\%}$: Pressure support ventilation 100% P$_{high}$; P$_{high}$ = spontaneous breaths at high continuous positive airway pressure.

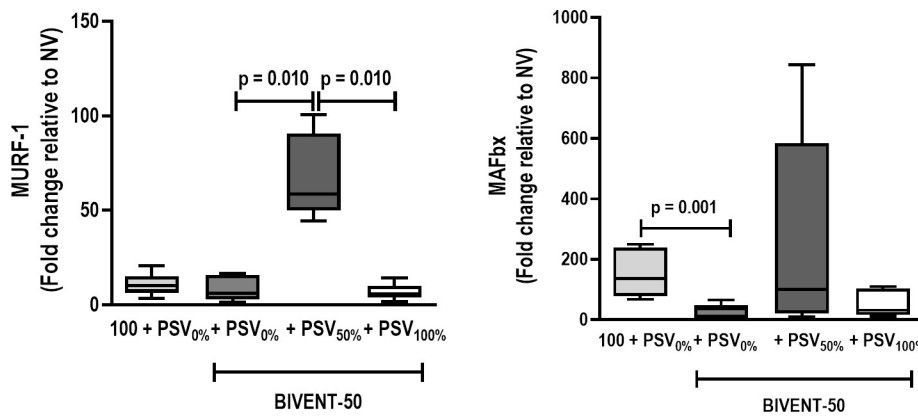

**Fig 7. Real-time polymerase chain reaction analysis of biological markers for proteolysis [muscle RING finger-1 (MuRF-1) and muscle atrophy F-box (MAFbx/atrogin-1)].** Box plots represent the median and interquartile range of 8 animals. Relative gene expression was calculated as a ratio of the average gene expression levels compared with the reference gene (*36B4*) and expressed as fold change relative to respective NV (nonventilated). Comparisons between BIVENT-100+PSV$_{0\%}$ and BIVENT-50+PSV0% groups were done by the Mann–Whitney *U* test (p<0.05). For comparisons within BIVENT-50 groups, the Kruskal-Wallis test with Dunn's post-hoc test was used (p<0.05).

mechanical ventilation strategies (second hit) were analyzed [30,31]. After the first hit, both the lung [32,33] and diaphragm [34] are more prone to injury. BIVENT is characterized by two levels (P$_{high}$ and P$_{low}$) of continuous positive airway pressure with unrestricted spontaneous breathing [17,18,35]. Additionally, BIVENT can be combined with PSV, as classically done in previous studies [18,36,37]. Adding PSV is expected to achieve a reduction in work of breathing [38] and increased alveolar recruitment. By gradually increasing the pressure support according to P$_{high}$ (0, 50%, and 100%) within BIVENT-50 groups, a "U-shaped" response was observed according to histological and molecular biology parameters: lung and diaphragm protection was observed in BIVENT-50+PSV$_{0\%}$ and BIVENT-50+PSV$_{100\%}$ groups, whereas BIVENT-50+PSV$_{50\%}$ impaired both lungs and the diaphragm.

## BIVENT at different controlled breaths without PSV (BIVENT-100+PSV$_{0\%}$ *vs* BIVENT-50+PSV$_{0\%}$)

BIVENT-100+PSV$_{0\%}$ was associated with a lower CV of V$_T$ than BIVENT-50+PSV$_{0\%}$. This can be explained by the higher number of mandatory cycles. The CV of V$_T$ achieved at BIVENT-50+PSV$_{0\%}$ was 32%, which may reduce lung damage [39,40]. Accordingly, the increased rate of spontaneous breathing at P$_{low}$ during BIVENT-50+PSV$_{0\%}$ was associated with reduced cumulative DAD score, mainly due to less alveolar collapse, less damage to epithelial/endothelial cells and basement membrane, and lower amphiregulin gene expression, which denotes less alveolar stretch [41]. Oxygenation did not differ between groups. This is consistent with the fact that oxygenation was associated with a balance between alveolar collapse and overdistension. Moreover, during assisted breathing, not only lung morphology but also regional perfusion distribution may play a relevant role in oxygenation [36].

By reducing the number of controlled cycles from BIVENT-100+PSV$_{0\%}$ to BIVENT-50+-PSV$_{0\%}$, spontaneous breathing cycles may occur, mainly at P$_{low}$ within a protective V$_T$ range. Both the appropriate degree of variability of respiratory pattern and better maintenance of respiratory muscle tone may improve recruitment and maintenance of airway patency through the modulation of different airway pressures and inspiratory times, ultimately maximizing lung recruitment and stabilization [40,42,43], without causing diaphragm injury. In this line, we may further infer that the maintenance of respiratory muscle tone during BIVENT-50

+$PSV_{0\%}$, but not in BIVENT-100+$PSV_{0\%}$, may have contributed to low diaphragm score and decreased expression of proteolysis markers. Accordingly, spontaneous breathing, compared to controlled mechanical ventilation, did not result in a significant decline in diaphragm protein synthesis [44], which corroborates our hypothesis.

## BIVENT-50 at different PSV (BIVENT-50+$PSV_{0\%}$, BIVENT-50+$PSV_{50\%}$, and BIVENT-50+$PSV_{100\%}$)

In BIVENT-50+$PSV_{50\%}$, compared to BIVENT-50+$PSV_{0\%}$, $V_T$ did not change; however, Ppeak,L was higher at $P_{high}$, reflecting vigorous efforts, which may have contributed to increase the level of $PTP_{min}$ and ΔPes. The $PTP_{min}$ was calculated as the integral of ΔPes over one minute and may better reflect inspiratory effort than esophageal pressure swing *per se*. Total respiratory rate did not differ among groups. Total ΔPes (variation of esophageal pressure during the inspiratory effort) was higher in BIVENT-50+$PSV_{50\%}$ compared to BIVENT-50+$PSV_{100\%}$, mainly due to the increase during PSV (assisted and spontaneous breaths). In BIVENT-50+-$PSV_{50\%}$, for the same airway pressure set on the ventilator, the higher the ΔPes at $P_{high}$ (7.3 ± 3.6 $cmH_2O$), the higher the Ppeak,L (22.1 ± 3.0 $cmH_2O$). On the other hand, when no pressure support was given (BIVENT-50+$PSV_{0\%}$), the lower the ΔPes at $P_{high}$ (4.4 ± 0.3 $cmH_2O$), the lower the Ppeak,L (15.3 ± 1.2 $cmH_2O$). We hypothesized that the increased expression of genes implicated in lung inflammation, extracellular matrix damage, and alveolar stretch in BIVENT-50+$PSV_{50\%}$ animals may be attributed to increased Ppeak,L and $PTP_{min}$ at $P_{high}$. Moreover, since BIVENT-50+PSV50% animals exhibited greater atelectasis and overdistension, the increase in Ppeak,L might also reflect a reduction in lung compliance. Increased inspiratory effort may lead an imbalanced diaphragm length-tension relationship [45] and thus culminate in diaphragmatic injury. In this line, Ppeak,L is an important driver of lung damage also when PSV (without BIVENT) is gradually reduced [38,46].

On the other hand, during BIVENT-50+$PSV_{0\%}$, animals showed inspiratory effort at spontaneous breathing mainly in the $P_{low}$ phase, which may protect the lungs against overdistension and triggering of biological markers. We may infer that the presence of spontaneous breathing activity at $P_{low}$ may mitigate VILI.

Animals tended to breathe spontaneously more at $P_{low}$ than at $P_{high}$ when the level of pressure support increased from 50 to 100%. In this line, the SERVO-i ventilator allows PSV breaths only at $P_{low}$ and not at $P_{high}$. Therefore, in BIVENT-50+$PSV_{50\%}$, 8 ± 1 (mean ± SD) spontaneous breaths occurred at $P_{high}$, whereas in BIVENT-50+$PSV_{100\%}$, 9 ± 2 spontaneous breaths occurred at $P_{low}$. In this context, if the level of pressure support is low, spontaneous breaths (assisted or not) are favored at higher lung volumes. On the other hand, if the level of pressure support is high (BIVENT-50+$PSV_{100\%}$), spontaneous breaths tend to be favored at lower lung volumes, resulting in less lung stretch and diaphragm injury, which is consistent with the literature [18,38]. BIVENT-50+$PSV_{100\%}$ seems to be the most promising ventilation mode.

## Possible clinical implications of study findings

The findings of the present study expand the knowledge based on assisted mechanical ventilation strategies by showing that, during BIVENT, both the frequency of controlled breaths and the levels of pressure support (0%, 50% and 100%) affect lung and diaphragm damage differently. In addition, lung injury was worse if the ventilator was set to promote spontaneous efforts at $P_{high}$ level, such as observed at BIVENT-50+$PSV_{50\%}$. When BIVENT is set with different mandatory (controlled) and spontaneous breaths (PSV-assisted or not), $PTP_{min}$ (as a surrogate of inspiratory effort) and Ppeak,L need to be measured during spontaneous breaths

at $P_{high}$ and $P_{low}$. In this line, animals ventilated at BIVENT-50+PSV$_{0\%}$ and BIVENT-50+-PSV$_{100\%}$ tended to breathe at lower pressures ($P_{low}$), whereas during BIVENT-50+PSV$_{50\%}$, they adapted at higher pressures ($P_{high}$), resulting in VILI and diaphragmatic damage. This reinforces the concept of the utility of esophageal pressure measurement at the bedside to optimize assisted breathing when targeted to minimize lung and diaphragm injury.

## Limitations

Some limitations of this study must be noted. First, an experimental model of mild pulmonary ARDS induced by intratracheal *E. coli* LPS instillation was used, which does not reproduce all features of human ARDS, does not apply to other degrees of ARDS severity, and is not representative of extrapulmonary ARDS. Second, the extent of alveolar permeability (measured by the protein content in bronchoalveolar lavage fluid) was not evaluated. Third, we chose not to ventilate healthy animals in order to avoid an overly large number of groups, and then increased the number of animals per group to maintain the power of the study. Finally, the ventilation period was limited to 1 hour, since longer periods of ventilation would have required infusion of additional fluids or even vasopressors to maintain MAP, which might have confounded the readouts. Therefore, we cannot guarantee that similar alterations would be maintained for longer periods. Nevertheless, 1 hour of mechanical ventilation was enough to observe molecular changes in key biological markers related to VILI and diaphragmatic proteolysis.

## Conclusions

In the ARDS model used herein, during BIVENT, the level of PSV and the phase of the respiratory cycle in which the inspiratory effort occurs affected lung and diaphragm damage. Lung injury was not influenced by the total values of inspiratory effort or transpulmonary pressure. Partitioning of these parameters in spontaneous breaths at $P_{low}$ and $P_{high}$ is required to minimize VILI.

## Supporting information

**S1 Table. Forward and reverse oligonucleotide sequences of target gene primers.**
(DOCX)

**S2 Table. Mean arterial pressure, amount of fluids infused, and arterial blood gases at timepoint BASELINE-ZEEP.**
(DOCX)

**S3 Table. Mean arterial pressure, amount of fluids infused, and arterial blood gases at timepoint FINAL.**
(DOCX)

**S4 Table. Respiratory parameters adjusted at the ventilator at BASELINE-ZEEP, INITIAL and FINAL.**
(DOCX)

**S1 File. Custom-made software written in LabVIEW and routine written in MATLAB for data analysis.**
(DOCX)

## Acknowledgments

The authors express our gratitude to Andre Benedito da Silva, B.Sc., Laboratory of Pulmonary Investigation, Carlos Chagas Filho Biophysics Institute, Federal University of Rio de Janeiro, Rio de Janeiro, Brazil, for animal care; Arlete Fernandes, B.Sc., Laboratory of Pulmonary Investigation, Carlos Chagas Filho Biophysics Institute, Federal University of Rio de Janeiro, Rio de Janeiro, Brazil, for her help with microscopy; Maíra Rezende Lima, M.Sc., Laboratory of Pulmonary Investigation, Carlos Chagas Filho Biophysics Institute, Federal University of Rio de Janeiro, Rio de Janeiro, Brazil, for her assistance in molecular biology analysis; Moira Elizabeth Schottler, B.A., Rio de Janeiro, Brazil, and Filippe Vasconcellos, B.A., São Paulo, Brazil, for their assistance in editing the manuscript; Ronir Raggio Luiz, Ph.D., Institute of Public Health Studies, Federal University of Rio de Janeiro, Rio de Janeiro, Brazil, for his help with statistics; and Maquet, Germany, for technical support.

## Author Contributions

**Conceptualization:** Alessandra F. Thompson, Marcelo G. de Abreu, Felipe Saddy, Paolo Pelosi, Pedro L. Silva, Patricia R. M. Rocco.

**Data curation:** Alessandra F. Thompson, Lillian Moraes, Soraia C. Abreu, Cynthia S. Samary, Felipe Saddy, Pedro L. Silva, Patricia R. M. Rocco.

**Formal analysis:** Alessandra F. Thompson, Lillian Moraes, Nazareth N. Rocha, Marcos V. S. Fernandes, Mariana A. Antunes, Soraia C. Abreu, Cintia L. Santos, Vera L. Capelozzi, Cynthia S. Samary, Patricia R. M. Rocco.

**Funding acquisition:** Patricia R. M. Rocco.

**Investigation:** Nazareth N. Rocha, Mariana A. Antunes, Cintia L. Santos, Vera L. Capelozzi, Cynthia S. Samary, Felipe Saddy, Paolo Pelosi, Pedro L. Silva, Patricia R. M. Rocco.

**Methodology:** Alessandra F. Thompson, Lillian Moraes, Nazareth N. Rocha, Marcos V. S. Fernandes, Mariana A. Antunes, Soraia C. Abreu, Cynthia S. Samary, Felipe Saddy, Pedro L. Silva.

**Project administration:** Cynthia S. Samary, Felipe Saddy, Pedro L. Silva, Patricia R. M. Rocco.

**Resources:** Soraia C. Abreu, Vera L. Capelozzi, Patricia R. M. Rocco.

**Software:** Cintia L. Santos, Cynthia S. Samary, Marcelo G. de Abreu, Pedro L. Silva.

**Supervision:** Mariana A. Antunes, Cynthia S. Samary, Paolo Pelosi, Pedro L. Silva, Patricia R. M. Rocco.

**Validation:** Lillian Moraes, Soraia C. Abreu, Cintia L. Santos, Vera L. Capelozzi, Cynthia S. Samary, Marcelo G. de Abreu, Felipe Saddy, Paolo Pelosi, Pedro L. Silva, Patricia R. M. Rocco.

**Visualization:** Soraia C. Abreu, Cintia L. Santos, Vera L. Capelozzi, Cynthia S. Samary, Marcelo G. de Abreu, Felipe Saddy, Paolo Pelosi, Pedro L. Silva, Patricia R. M. Rocco.

**Writing – original draft:** Alessandra F. Thompson, Marcelo G. de Abreu, Paolo Pelosi, Pedro L. Silva, Patricia R. M. Rocco.

**Writing – review & editing:** Alessandra F. Thompson, Lillian Moraes, Nazareth N. Rocha, Marcos V. S. Fernandes, Mariana A. Antunes, Soraia C. Abreu, Cintia L. Santos, Vera L. Capelozzi, Cynthia S. Samary, Marcelo G. de Abreu, Felipe Saddy, Paolo Pelosi, Pedro L. Silva, Patricia R. M. Rocco.

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
