## [Decision Letter · Decision Letter 0]

4 Mar 2021

PONE-D-20-40795

Impact of different frequencies of controlled breath and pressure-support levels during biphasic positive airway pressure ventilation on the lung and diaphragm in experimental mild acute respiratory distress syndrome

PLOS ONE

Dear Dr. Rocco,

Thank you for submitting your manuscript to PLOS ONE. After careful consideration, we feel that it has merit but does not fully meet PLOS ONE’s publication criteria as it currently stands. Therefore, we invite you to submit a revised version of the manuscript that addresses the points raised during the review process.

Please note that relevant criticism was raised regarding the methodology of the presented experiments: a bias in that area is a reason for rejection due to Plos One standards. Therefore, please carefully respond to all the comments prensented by the reviewers.

Regarding the presented results, the strongest signal appears related to the bipap50-psv50 group. Looking at the difference in table 1, 8 breaths were at pHigh in the bipap50-psv50 group (no breaths at plow: why?), while 9 breaths at pLow for the bipap50-psv100 group (no breaths at phigh: why?). A different respiratory pattern could be the reason leading to sili? Why the psv 50 rats breathe at pHigh and the psv100 at pLow?

Another point that needs to be clarified is the peak transpulmonary pressure: the bipap50-psv50 group showed 22 at pHigh vs.15 in the bipap50-psv0 group; how do you explain? I expect that it should be the same. Is it possible that 22 is due to PSV over the pHigh (although, based on the methods section, that should not be the case: pHigh sholud refer to the spontaneous non PSV breaths at cpapHigh). If not, did you discriminate if PSV breaths occurred at pLow vs. pHigh? A triggered PSV cycle above pHigh might lead to overdistention and stretch, possibly explaining some of your results.

We look forward to receiving your revised manuscript.

Kind regards,

Andrea Coppadoro

Academic Editor

PLOS ONE

2. Please provide the source of the animals used in your study.

3. In order to meet our data availability and reproducibility criteria, please provide your custom-made software written in LabVIEW and your routine written in MATLAB for data analysis. You may either provide copies of the code as supplementary files, or you can provide a link to a permanent URL.

Reviewers' comments:

Reviewer's Responses to Questions

**Comments to the Author**

1. Is the manuscript technically sound, and do the data support the conclusions?

Reviewer #1: Yes

Reviewer #2: Yes

Reviewer #3: Partly

2. Has the statistical analysis been performed appropriately and rigorously? 

Reviewer #1: Yes

Reviewer #2: Yes

Reviewer #3: Yes

3. Have the authors made all data underlying the findings in their manuscript fully available?

Reviewer #1: Yes

Reviewer #2: Yes

Reviewer #3: Yes

4. Is the manuscript presented in an intelligible fashion and written in standard English?

Reviewer #1: Yes

Reviewer #2: Yes

Reviewer #3: Yes

5. Review Comments to the Author

Reviewer #1: COMMENTS TO AUTHORS

The authors hypothesized that lung and diaphragm injury may be altered by levels of pressure support and frequency of controlled-mechanical breath during Bilevel ventilation. By using mild lung injury model, the authors found that 1) lung injury was less when spontaneous effort was facilitated by decreasing mandatory mechanical breath during Bilevel ventilation; 2) but lung injury was deteriorated when spontaneous effort assisted by pressure support occurred on a top of Phigh level. The reviewer found that the current study was intriguing and covered a hot topics regarding how physicians facilitated spontaneous breathing in lung injury. But on the other hand, it was difficult to grasp what the main message was from the current version of manuscript due to numbers of groups.

# message

The reviewer thinks the current study has two messages; first, lung injury was decreased when spontaneous effort was facilitated during Plow by decreasing mandatory mechanical breath during Bilevel ventilation (BIVENT100+PS0 vs. BIVENT50+PS0), second, lung injury was deteriorated if ventilatory settings were manipulated to promote spontaneous effort assisted by pressure support on a top of Phigh level (BIVENT50+PS50). Especially 2nd message is important and should be stressed in a text. Thus, the reviewer suggests the authors to clarify this in conclusion and discussion, in order to let the message more straightforward. Supplemental figure-1 showing sample waveforms indeed helps readers to grasp the message so that this should be implemented in main text.

# potential mechanism

What was surprising to me is that PS-assisted spontaneous effort was observed only 8/mins on a top of Phigh (BIVENT50+PS50), but injury was significantly different from others. How was that possible by such a low respiratory rate occurring during Phigh?

Reviewer #2: In this study, Thompson et al evaluated the lung and diaphragm injury in an experimental model of E.coli induced ARDS, followed by different BIVENT. The authors stated that the frequency of controlled breaths and the PSV level during BIVENT can affect lung and diaphragm damage.

In my opinion this is a very interesting study, that could help to test different mechanical ventilation strategies in ARDS patients. The present study is well planned and well written. For these reasons I have only minor comments

- In the results, there is no mention about the mortality of the animals, during the 24 hours after E.coli administration. Is it possible to add this aspect?

- Among the experimental groups the authors did not consider a group of rat only mechanically ventilated without LPS insult. Please explain the reason of this choice.

- I have only a doubt regarding the time of ventilation. I think (looking to available literature) that only 1 hour of mechanical ventilation is not sufficient to induce a diaphragm injury and a structural disorganization. Please hypothesize a possible explanation about this aspect.

- No differences in terms of oxygenation were found among the experimental groups. Was this result expected?

- In the text, the BIVENT-50-PSV100% was not deeply analyzed. In the Discussion, in “Comparisons across the BIVENT-50 groups” section, the focus was about the comparison between PSV0 versus PSV50. The authors could give more attention on PSV100 results and comparisons versus the other group, since the BIVENT-50-PSV100 seems to be the most promising ventilation method.

Reviewer #3: Thompson AF. et al. reported the effect of Bilevel at 2 set up of RR and with or without PS (i.e. BILEVEL 50) on macroscopic and microscopic variables of lung and diaphragm injury in a preclinical investigation using a rodent model of lung injury by IT instillation of E. Coli LPS.

The primary aim of the study is based on a sample size justified to evaluate the difference in alveolar collapse (i.e. 1 of the 3 items used to estimate DAD) among Bilevel settings using a higher or a lower fixed RR (i.e. 100 versus 50, respectively) and with no PS.

The authors further evaluate the presence of differences on the respiratory parameters only in the lower RR group of BILEVEL (i.e. 50 bpm) according to different levels of PS.

Although the work is of potential clinical interest at bedside, I think that some bias exists in both the animal model and in the interpretation of the study results that does not clearly stick to the study findings.

1. At first, the authors refer to an animal model of mild ARDS. I am not sure that this is correct. According to the criteria of Berlin, PEEP must be included to characterize the severity of ARDS – at the “initial time point” the authors state that PF was lower of 300 in all groups – however, as reported in the methods section – PEEP was 0 at baseline before randomization – this does not guarantee that levels of PF are below 300 in the presence of some level of PEEP at baseline. Furthermore, 1 h of ventilation with PEEP=5 cmH2O brings the PF ratio up to an average value way above 300 (FINAL timepoint) in all groups – and this further confirms that the definition of ARDS is not accurate. Data of gas exchange at baseline should be reported.

2. Quantification of TV of 6 mL/Kg – I have a similar thought as previously observed -considering that the quantification of a TV = 6 ml/kg would report a different driving pressure at ZEEP compared to a DP estimated at a level of PEEP=5 cmH2O – this suggests that the levels of PSV used after randomization to achieve 6 mL/kg was based on an estimation of DP performed at ZEEP and not at PEEP=5 – I expect a different DP for the same TV at ZEEP versus PEEP 5 because of a different position on the PV curve. The authors should report average levels of PSV used in the different groups and the average level of Phigh of the BILEVEL in all groups.

3. It is not clear what the authors mean about the following sentence: “Flow trigger sensitivity was adjusted for adequate inspiratory effort, according to esophageal pressure (Pes) decay”? Did the authors use a fixed flow trigger in all the PSV experiments, didn’t they? The use of different thresholds of flow trigger may make unreliable the study findings as it means that this variable was not kept the same among different PSV experiments.

4. Furthermore, the different PSV groups showed a total RR – despite not significantly different I assume because of the low sample size among the groups - ranging from an average of 81 (PSV 50%) to 113 bpm (PSV 100%) with an increasing level of PSV – which is kind of counterintuitive…I can’t buy for what I see this U-shaped concept.. As first, I would ask whether the level of sedation was kept constant among the different PSV steps or not as in a study previously published by the same investigators (doi.org/10.1371/journal.pone.0246891). Actually, I would expect a lower RR in the PSV100% group in the absence of brain injury.. Furthermore, this makes me uncomfortable about the calculation of the pressure–time product per minute (PTPmin) that was calculated as the integral of ΔPes over one minute – this is, certainly, affected by a different RR among the study groups – so I am not sure if the difference among groups in PTP is because of the animal respiratory effort or because of the different RR. Looking at the PTP of the 2 groups on BILEVEL 100 versus 50 and no PSV – it seems unlikely that the PTP in the BILEVEL 50 and no PSV does not differ compared to the BILEVEL 100 and no PSV in the presence of a 50% decrease of fixed breaths. Levels of Pes should be reported among the study groups.

5. Variability of Vt is quite harsh to interpret as at PSV0% versus PSV 100% CV of TV is basically the same

6. About the DAD score, I am not sure to understand such lower levels of edema and collapse (i.e. media of 2) and a median of 5 for overdistension in the presence of quite low levels of mean lung pressure (i.e. mean value of 4.1) in the PSV100% BILEVEL50

7. Significance was established at p<0.05 – was a two-sided p-value, is it correct? Please add this information.

8. Table 1: Airlow > typo, change it into airflow

9. Discussion: “Therefore, there is a certain threshold of PSV in BIVENT-50 that may yield a continuous excessive stress.” It is a speculation please remove it – the study wasn’t powered to assess a difference in stress among the groups - stress was not the primary study aim – furthermore the data in BILEVEL 50 PSV 50% shows a higher Pmean,L - although not significant compared to other groups - a lower variation of TV and a lower RR. However, in figure 2, despite a pretty low RR in BILEVEL 50 PSV 50%, the swings of Pes were quite limited compared to PSV0% and even PSV100% - this is quite a surprise to me looking at the PTP - again any difference in the assisted flow trigger or sedation?

6. PLOS authors have the option to publish the peer review history of their article (what does this mean?). If published, this will include your full peer review and any attached files.

Reviewer #1: No

Reviewer #2: No

Reviewer #3: No

---

## [Author Response · Author response to Decision Letter 0]

18 May 2021

Response to Editor´s Comments

Regarding the presented results, the strongest signal appears related to the bipap50-psv50 group. Looking at the difference in table 1, 8 breaths were at pHigh in the bipap50-psv50 group (no breaths at plow: why?), while 9 breaths at pLow for the bipap50-psv100 group (no breaths at phigh: why?). A different respiratory pattern could be the reason leading to sili? Why the psv 50 rats breathe at pHigh and the psv100 at pLow?

Response: We thank the Editor for this important observation. Indeed, by increasing the level of pressure support from 50% to 100%, animals tend to breathe spontaneously more at Plow than at Phigh. The SERVO-i ventilator allows PSV breaths only at Plow and not at Phigh. Therefore, in BIVENT-50+PSV50%, 8 ± 1 (mean ± SD) spontaneous breaths occurred at Phigh, whereas 9 ± 2 spontaneous breaths occurred at Plow in BIVENT-50+PSV100%. In this context, if the level of pressure support is low (BIVENT-50+PSV50%), spontaneous breaths (PSV assisted or not) are favored at higher lung volumes. On the other hand, if the level of pressure support is high (BIVENT-50+PSV100%), breaths tend to be favored at lower lung volumes, resulting in less lung stretch and diaphragm injury, as described in the literature. We have better specified and discussed this observation in the revised manuscript.

We also agree with the Editor that different respiratory patterns as well as spontaneous breaths at Plow and Phigh may affect VILI. In this line, the expression of genes associated with lung inflammation, epithelial and endothelial cell damage, extracellular matrix damage, and diaphragmatic injury were greater at BIVENT-50+PSV50%, in which spontaneous breaths occurred at Phigh, resulting in increased peak transpulmonary pressure (Ppeak,L) and pressure–time product per minute (PTPmin), which may have favored greater atelectasis and overdistension, thus worsening VILI. In contrast, an increased rate of spontaneous breaths at Plow did not promote lung injury. In short, total values of PTPmin (inspiratory effort) and Ppeak,L (transpulmonary pressure) did not contribute towards reduction of VILI; however, partitioning of these parameters between spontaneous breaths at Plow and Phigh is required during BIVENT to optimize ventilator settings.

In BIVENT-50+PSV50%, compared to BIVENT-50+PSV0%, VT did not change; however, Ppeak,L was higher at Phigh, reflecting vigorous efforts, which may have contributed to increase the level of PTPmin and �Pes. The PTPmin was calculated as the integral of ΔPes over one minute and may better reflect inspiratory effort than esophageal pressure swing per se. The total respiratory rate did not differ among groups. Total �Pes (variation of esophageal pressure during the inspiratory effort) was higher in BIVENT-50+PSV50% compared to BIVENT-50+PSV100%, mainly due to the increase during PSV (assisted and spontaneous breaths). In BIVENT-50+PSV50%, for the same airway pressure set on the ventilator, the higher the �Pes at Phigh (7.3 ± 3.6 cmH2O), the higher the Ppeak,L (22.1 ± 3.0 cmH2O). On the other hand, when no pressure support was given (BIVENT-50+PSV0%), the lower the �Pes at Phigh (4.4 ± 0.3 cmH2O), the lower the Ppeak,L (15.3 ± 1.2 cmH2O). We hypothesized that the increased expression of genes implicated in lung inflammation, extracellular matrix damage, and alveolar stretch in BIVENT-50+PSV50% animals may be attributed to increased Ppeak,L and PTPmin at Phigh. We have modified the Discussion to clarify this issue. 

Another point that needs to be clarified is the peak transpulmonary pressure (Ppeak,L): the bipap50-psv50 group showed 22 at pHigh vs.15 in the bipap50-psv0 group; how do you explain? I expect that it should be the same. Is it possible that 22 is due to PSV over the pHigh (although, based on the methods section, that should not be the case: pHigh sholud refer to the spontaneous non PSV breaths at cpapHigh). If not, did you discriminate if PSV breaths occurred at pLow vs. pHigh? A triggered PSV cycle above pHigh might lead to overdistention and stretch, possibly explaining some of your results.

Response: We agree with the Editor’s comments. As described in the Method section, Phigh (high airway pressure) reflects spontaneous breath cycles without PSV. Even though there were negative Pes variations at Phigh, this was not followed by ventilator cycling, since SERVO-I ventilator (MAQUET, Solna, Sweden) enables PSV only during Plow. We have now better explained this point in the Method section of the revised manuscript. 

We have now added the �Pes in the Table 1, as requested by one Reviewer. At BIVENT-50+PSV50%, Ppeak,L increased as well as PTPmin at Phigh, which may reflect excessive inspiratory effort. The PTPmin was calculated as the integral of ΔPes over one minute and may better reflect inspiratory effort than esophageal pressure decay per se. Nevertheless, we have now presented the maximum ΔPes decay in order to better reflect the lung stretch caused by spontaneous effort. Moreover, both components of PTPmin, i.e., RR and ΔPes, are now shown separately. Total RR did not differ among groups. On the other hand, total �Pes, which reflects the variation of esophageal pressure during the inspiratory effort, was higher in BIVENT-50+PSV50% than BIVENT-50+PSV100%. In BIVENT-50+PSV50%, total �Pes was highly influenced by �Pes observed during PSV cycles. As a consequence, the higher the �Pes at Phigh (7.3 ± 3.6 cmH2O) for the same airway pressure set on the ventilator, the higher Ppeak,L will be, as observed in BIVENT-50+PSV50% (22.1 ± 3.0 cmH2O). On the other hand, when no pressure support was given, the lower the �Pes at Phigh (4.4 ± 0.3 cmH2O) for the same airway pressure set on the ventilator, the lower Ppeak,L will be (15.3 ± 1.2 cmH2O). 

These findings have important clinical implications. Lung injury was worse if the ventilator was set to promote spontaneous efforts at Phigh, such as observed at BIVENT-50+PSV50%. When BIVENT is set with different mandatory (controlled) and spontaneous breaths (PSV assisted or not), PTPmin (as a surrogate of inspiratory effort) and Ppeak,L at Phigh and Plow during spontaneous breath should be measured by esophageal pressure in order to optimize and individualize these ventilatory settings, thus reducing the risk of lung and diaphragm injury. In addition, BIVENT-50+PSV50% was associated with higher atelectasis and overdistension; thus, the increase in Ppeak,L and �Pes might also reflect reduction in lung compliance. We have better explained these important and critical issues in the revised version of the manuscript.

Response to Reviewers´ Comments

Reviewer#1

The authors hypothesized that lung and diaphragm injury may be altered by levels of pressure support and frequency of controlled-mechanical breath during Bilevel ventilation. By using mild lung injury model, the authors found that 1) lung injury was less when spontaneous effort was facilitated by decreasing mandatory mechanical breath during Bilevel ventilation; 2) but lung injury was deteriorated when spontaneous effort assisted by pressure support occurred on a top of Phigh level. The reviewer found that the current study was intriguing and covered a hot topics regarding how physicians facilitated spontaneous breathing in lung injury. But on the other hand, it was difficult to grasp what the main message was from the current version of manuscript due to numbers of groups.

Response: We thank the Reviewer for these positive comments. In the revised version of the manuscript, several modifications were done to clarify the main message of the study. 

# message

The reviewer thinks the current study has two messages; first, lung injury was decreased when spontaneous effort was facilitated during Plow by decreasing mandatory mechanical breath during Bilevel ventilation (BIVENT100+PS0 vs. BIVENT50+PS0), second, lung injury was deteriorated if ventilatory settings were manipulated to promote spontaneous effort assisted by pressure support on a top of Phigh level (BIVENT50+PS50). Especially 2nd message is important and should be stressed in a text. Thus, the reviewer suggests the authors to clarify this in conclusion and discussion, in order to let the message more straightforward. Supplemental figure-1 showing sample waveforms indeed helps readers to grasp the message so that this should be implemented in main text.

Response: We thank the Reviewer for these important suggestions, which have now been incorporated to the main text. Figure 1 has been moved to the main manuscript. 

Additionally, we would like to clarify that at Phigh there was no PSV, only spontaneous breath cycles. Even though there were negative Pes variations at Phigh, this was not followed by ventilator cycling since SERVO-I ventilator (MAQUET, Solna, Sweden) enables PSV only during Plow phase. We have now better explained this point in the Method section.

As requested by the Reviewer, we have now added a sentence in the “possible clinical implications of study findings” section, as follows: “Lung injury was worse if the ventilator was set to promote spontaneous efforts at Phigh level, such as observed at BIVENT-50+PSV50%. When BIVENT is set with different mandatory (controlled) and spontaneous breaths (PSV assisted or not), PTPmin (as a surrogate of inspiratory effort) and Ppeak,L need to be measured during spontaneous breaths at Phigh and Plow. In this line, animals ventilated at BIVENT-50+PSV0% and BIVENT-50+PSV100% tended to breathe at lower pressures (Plow), whereas during BIVENT-50+PSV50%, they adapted at higher pressures (Phigh), resulting in VILI and diaphragmatic damage. This reinforces the concept of the utility of esophageal pressure measurement at the bedside to optimize assisted breathing when targeted to minimize lung and diaphragm injury. 

Furthermore, we better clarified our conclusion: “In the ARDS model used herein, during BIVENT, the level of PSV and the phase of the respiratory cycle in which the inspiratory effort occurs affected lung and diaphragm damage. Lung injury was not influenced by the total values of inspiratory effort or transpulmonary pressure. Partitioning of these parameters in spontaneous breaths at Plow and Phigh is required to minimize VILI”.

# potential mechanism

What was surprising to me is that PS-assisted spontaneous effort was observed only 8/mins on a top of Phigh (BIVENT50+PS50), but injury was significantly different from others. How was that possible by such a low respiratory rate occurring during Phigh?

Response: We thank the Reviewer for this important question. Indeed, by increasing the level of pressure support from 50% to 100%, animals tend to breathe spontaneously more at Plow than at Phigh. The SERVO-i ventilator allows PSV breaths only at Plow and not at Phigh. Therefore, in BIVENT-50+PSV50%, 8 ± 1 (mean ± SD) spontaneous breaths occurred at Phigh, whereas 9 ± 2 spontaneous breaths occurred at Plow in BIVENT-50+PSV100%. In this context, if the level of pressure support is low (BIVENT-50+PSV50%), spontaneous breaths (PSV assisted or not) are favored at higher lung volumes and pressures. On the other hand, if the level of pressure support is high (BIVENT-50+PSV100%), breaths tend to be favored at lower lung volumes and pressures, resulting in less lung stretch and diaphragm injury, as described in the literature.

These findings have important clinical implications. Lung injury was worse if the ventilator was set to promote spontaneous efforts at Phigh, such as observed at BIVENT-50+PSV50%. When BIVENT is set with different mandatory (controlled) and spontaneous breaths (PSV assisted or not), PTPmin (as a surrogate of inspiratory effort) and Ppeak,L at Phigh and Plow during spontaneous breath should be measured by esophageal pressure in order to optimize and individualize these ventilatory settings, thus reducing the risk of lung and diaphragm injury. In addition, BIVENT-50+PSV50% was associated with higher atelectasis and overdistension; thus, the increase in Ppeak,L and �Pes might also reflect reduction in lung compliance. We have better explained these important and critical issues in the revised version of the manuscript.

Reviewer #2 

In this study, Thompson et al evaluated the lung and diaphragm injury in an experimental model of E.coli induced ARDS, followed by different BIVENT. The authors stated that the frequency of controlled breaths and the PSV level during BIVENT can affect lung and diaphragm damage.

In my opinion this is a very interesting study, that could help to test different mechanical ventilation strategies in ARDS patients. The present study is well planned and well written. For these reasons I have only minor comments

Response: We thank the Reviewer for these positive comments. 

- In the results, there is no mention about the mortality of the animals, during the 24 hours after E.coli administration. Is it possible to add this aspect?

Response: We thank the Reviewer for this important question. Two animals died due to hemodynamic compromise during the pilot phase. We have now included this information in the revised manuscript.

- Among the experimental groups the authors did not consider a group of rat only mechanically ventilated without LPS insult. Please explain the reason of this choice.

Response: We chose not to ventilate healthy animals in order to avoid an overly large number of groups and then increased the number of animals per group to maintain the power of the study as well as to focus on the main hypothesis. The Limitation section has been modified to better clarify this issue.

- I have only a doubt regarding the time of ventilation. I think (looking to available literature) that only 1 hour of mechanical ventilation is not sufficient to induce a diaphragm injury and a structural disorganization. Please hypothesize a possible explanation about this aspect.

Response: We agree with the Reviewer that 1 hour of mechanical ventilation is not sufficient to observe structural alterations in the diaphragm using light microscopy. However, such a short period in small animals does enable us to evaluate early changes in diaphragm using molecular biology (RT-PCR) and electron microscopy. Therefore, we measured the gene expression (mRNA levels) of markers associated with proteolysis: muscle RING finger-1 (MuRF-1) and muscle atrophy F-box (MAFbx/atrogin-1). According to previous studies from our group (Cruz, PlosOne 2021), 1 hour of mechanical ventilation was sufficient to detect diaphragm changes in TNF-alpha levels associated with different partial ventilatory support modes. It should be pointed out that the biological stimulus occurred 24 hours before, with LPS instillation. Thus, LPS animals are more prone to injury compared to healthy animals being subjected solely to mechanical ventilation. In relation to electron microscopy, we mainly focused on diaphragmatic mitochondrial damage, sarcomere disarrangement, and Z-disc edema. According to previous studies from our group (Saddy, Critical Care 2013), diaphragm injury was observed as well as the presence of vacuoles. In short, in small animals, the first hit is the administration of LPS 24 h before mechanical ventilation, whereas the second hit is the mechanical ventilation strategy itself. After the first hit, both the lung (Felix et al., Anesthesiology, 2019; Rocco & Marini, ICM 2021) and diaphragm (Shimada et al., Immunity, 2012) are primed and thus more prone to injury. The manuscript has been modified accordingly.

- No differences in terms of oxygenation were found among the experimental groups. Was this result expected?

Response: In our first paper (Saddy et al., Intensive Care Med, 2010) using this ventilatory mode, changes in oxygenation were observed when assisted ventilation modes were compared with PCV. Oxygenation improvement was associated with reduced atelectasis. However, when comparisons were done between assisted ventilation modes, no significant differences were observed in oxygenation (Saddy et al., Crit Care, 2013, Cruz et al. PlosOne 2021); this is consistent with the fact that oxygenation was associated with a balance between alveolar collapse and overdistension. Moreover, during assisted breathing, not only lung morphology but also regional perfusion distribution may play a relevant role in oxygenation (Carvalho et al., JAP 2011). The Discussion section has been modified to clarify this issue.

- In the text, the BIVENT-50-PSV100% was not deeply analyzed. In the Discussion, in “Comparisons across the BIVENT-50 groups” section, the focus was about the comparison between PSV0 versus PSV50. The authors could give more attention on PSV100 results and comparisons versus the other group, since the BIVENT-50-PSV100 seems to be the most promising ventilation method.

Response: We thank the Reviewer for this comment. The manuscript has been modified accordingly: “if the level of pressure support is high (BIVENT-50+PSV100%), spontaneous breaths tend to be favored at lower lung volumes, resulting in less lung stretch and diaphragm injury, which is consistent with the literature. BIVENT-50+PSV100% seems to be the most promising ventilation mode.”

These results have important clinical implications. “Lung injury was worse if the ventilator was set to promote spontaneous efforts at Phigh level, such as observed at BIVENT-50+PSV50%. When BIVENT is set with different mandatory (controlled) and spontaneous breaths (PSV assisted or not), PTPmin (as a surrogate of inspiratory effort) and Ppeak,L need to be measured during spontaneous breaths at Phigh and Plow. In this line, animals ventilated at BIVENT-50+PSV0% and BIVENT-50+PSV100% tended to breathe at lower pressures (Plow), whereas during BIVENT-50+PSV50%, they adapted at higher pressures (Phigh), resulting in VILI and diaphragmatic damage. This reinforces the concept of the utility of esophageal pressure measurement at the bedside to optimize assisted breathing when targeted to minimize lung and diaphragm injury.” 

We have also clarified the conclusion: “During BIVENT, the level of PSV and the phase of the respiratory cycle in which the inspiratory effort occurs affected lung and diaphragm damage. Lung injury was not influenced by the total values of inspiratory effort or transpulmonary pressure. Partitioning of these parameters in spontaneous breaths at Plow and Phigh is required to minimize VILI.

 

Reviewer #3 

Thompson AF. et al. reported the effect of Bilevel at 2 set up of RR and with or without PS (i.e. BILEVEL 50) on macroscopic and microscopic variables of lung and diaphragm injury in a preclinical investigation using a rodent model of lung injury by IT instillation of E. Coli LPS.

The primary aim of the study is based on a sample size justified to evaluate the difference in alveolar collapse (i.e. 1 of the 3 items used to estimate DAD) among Bilevel settings using a higher or a lower fixed RR (i.e. 100 versus 50, respectively) and with no PS.

The authors further evaluate the presence of differences on the respiratory parameters only in the lower RR group of BILEVEL (i.e. 50 bpm) according to different levels of PS.

Although the work is of potential clinical interest at bedside, I think that some bias exists in both the animal model and in the interpretation of the study results that does not clearly stick to the study findings.

Response: We thank the Reviewer for these comments, and modifications will be incorporated to clarify the main message.

1. At first, the authors refer to an animal model of mild ARDS. I am not sure that this is correct. According to the criteria of Berlin, PEEP must be included to characterize the severity of ARDS – at the “initial time point” the authors state that PF was lower of 300 in all groups – however, as reported in the methods section – PEEP was 0 at baseline before randomization – this does not guarantee that levels of PF are below 300 in the presence of some level of PEEP at baseline. Furthermore, 1 h of ventilation with PEEP=5 cmH2O brings the PF ratio up to an average value way above 300 (FINAL timepoint) in all groups – and this further confirms that the definition of ARDS is not accurate. Data of gas exchange at baseline should be reported.

Response: We thank the Reviewer for this important question. We have now included the gas-exchange at BASELINE ZEEP as requested in a new supplemental table. It can be noted that all groups presented a mean PaO2/FiO2 <300 mmHg.

We understand the Reviewer’s concern regarding experimental ARDS vs. clinical ARDS criteria. Nevertheless, experimental mild ARDS induced by intratracheal instillation of E. coli lipopolysaccharide reproduces several features of mild human ARDS (Matute-Bello G, 2011). According to the ATS Acute Lung Injury in Animals Study Group (Matute-Bello G, 2011), in small animals, changes in lung histology associated with increased neutrophil infiltration are more closely associated with lung damage than evaluation of oxygenation. Thus, in experimental rather than clinical settings, oxygenation is not as useful a parameter to evaluate the degree of lung injury. In the revised manuscript, the Discussion section has been modified accordingly. 

The levels of PSV used after randomization to achieve 6 mL/kg was based on an estimation of DP performed at ZEEP and not at PEEP=5 – I expect a different DP for the same TV at ZEEP versus PEEP 5 because of a different position on the PV curve. The authors should report average levels of PSV used in the different groups and the average level of Phigh of the BILEVEL in all groups.

Response: We thank the Reviewer for this question. The levels of PSV used after randomization to achieve 6 mL/kg were based on the estimation of �P performed at PEEP=5 cmH2O. Indeed, immediately after PEEP was added (INITIAL) no significant changes in �P was observed, however, a progressive increase in �P was found at the end of the experiment in BIVENT-50+PSV50%. We have now reported the range of adjusted PSV as well as the range of adjusted Phigh (at BASELINE ZEEP, INITIAL and FINAL) of the BIVENT in a new supplemental table.

3. It is not clear what the authors mean about the following sentence: “Flow trigger sensitivity was adjusted for adequate inspiratory effort, according to esophageal pressure (Pes) decay”? Did the authors use a fixed flow trigger in all the PSV experiments, didn’t they? The use of different thresholds of flow trigger may make unreliable the study findings as it means that this variable was not kept the same among different PSV experiments.

Response: We thank the Reviewer for this question. We have now better described how the flow trigger sensitivity was adjusted. Flow trigger sensitivity was adjusted at BASELINE-PEEP (INITIAL) for adequate inspiratory effort, according to esophageal pressure variation (�Pes). No additional changes to flow trigger sensitivity were made at any point during the experiment. 

4. Furthermore, the different PSV groups showed a total RR – despite not significantly different I assume because of the low sample size among the groups - ranging from an average of 81 (PSV 50%) to 113 bpm (PSV 100%) with an increasing level of PSV – which is kind of counterintuitive…I can’t buy for what I see this U-shaped concept.. As first, I would ask whether the level of sedation was kept constant among the different PSV steps or not as in a study previously published by the same investigators (doi.org/10.1371/journal.pone.0246891). Actually, I would expect a lower RR in the PSV100% group in the absence of brain injury. 

Response: Indeed, we did not find difference in total RR among groups. Moreover, no significant changes were observed between BIVENT-50+PSV50% and BIVENT-50+PSV100%. During spontaneous breathing, the level of anesthesia was assessed by evaluating pupil size, position, and response to light, position of the nictating membrane, and movement in response to tail stimulation. Sedation and anesthesia were kept constant throughout the experiment. Method section has been modified to better clarify this issue. BIVENT-50+PSV50% compared to BIVENT-50+PSV100% resulted in spontaneous breaths at Phigh, leading to increased lung and diaphragm damage. 

Furthermore, this makes me uncomfortable about the calculation of the pressure–time product per minute (PTPmin) that was calculated as the integral of ΔPes over one minute – this is, certainly, affected by a different RR among the study groups – so I am not sure if the difference among groups in PTP is because of the animal respiratory effort or because of the different RR. Looking at the PTP of the 2 groups on BILEVEL 100 versus 50 and no PSV – it seems unlikely that the PTP in the BILEVEL 50 and no PSV does not differ compared to the BILEVEL 100 and no PSV in the presence of a 50% decrease of fixed breaths. Levels of Pes should be reported among the study groups.

Response: We thank the Reviewer for this comment. 

We did not detect differences in total RR; however, we found differences concerning compensatory increase in RR in different phases of BIVENT, whether at Thigh or Tlow. If total RR would be a determinant of total PTPmin, they would show similar behavior among different groups. We can observe that total PTPmin was lower in BIVENT-50+PSV100% compared to 0% and 50% of PSV even at similar total RR. This infers lower overall inspiratory effort in the BIVENT-50+PSV100% group. The PTPmin was calculated as the integral of ΔPes over one minute and may better reflect inspiratory effort than esophageal pressure decay per se. Nevertheless, we have now shown the maximum ΔPes decay in order to better reflect the lung stretch caused by spontaneous effort. Both components of PTPmin, i.e., RR and ΔPes, are now shown separately. Total RR did not differ among groups. On the other hand, total �Pes, which reflects the variation of esophageal pressure during the inspiratory effort, was higher in BIVENT-50+PSV50% than BIVENT-50+PSV100%. In BIVENT-50+PSV50%, total �Pes was highly influenced by �Pes observed during PSV cycles since they were adjusted to half the pressure support compared to full support in BIVENT-50+PSV100%. We have explained this issue more clearly in the revised manuscript. 

5. Variability of Vt is quite harsh to interpret as at PSV0% versus PSV 100% CV of TV is basically the same

Response: We computed the variability of VT as a possible explanation for adequacy of natural (or most) neurological drive. We observed that, at 0% or 100% of PSV, the animals maintained physiological CV of VT, which is associated with positive outcomes, according to previous literature (Thammanomai et al. J. Appl Physiol, 2008). 

6. About the DAD score, I am not sure to understand such lower levels of edema and collapse (i.e. media of 2) and a median of 5 for overdistension in the presence of quite low levels of mean lung pressure (i.e. mean value of 4.1) in the PSV100% BILEVEL50

Response: We thank the Reviewer for this question. These levels of interstitial edema, alveolar collapse, and overdistension are in accordance with previous studies from our group using mild lung damage induced by endotoxin (Felix et al., Anesthesiology, 2019). Moreover, these morphological changes may explain the low levels of mean transpulmonary pressure.

7. Significance was established at p<0.05 – was a two-sided p-value, is it correct? Please add this information.

Response: Yes, a two-sided p-value. We have now added this information to the manuscript.

8. Table 1: Airlow > typo, change it into airflow

Response: We apologize for this mistake. We have now corrected it. 

9. Discussion: “Therefore, there is a certain threshold of PSV in BIVENT-50 that may yield a continuous excessive stress.” It is a speculation please remove it – the study wasn’t powered to assess a difference in stress among the groups - stress was not the primary study aim – furthermore the data in BILEVEL 50 PSV 50% shows a higher Pmean,L - although not significant compared to other groups - a lower variation of TV and a lower RR. However, in figure 2, despite a pretty low RR in BILEVEL 50 PSV 50%, the swings of Pes were quite limited compared to PSV0% and even PSV100% - this is quite a surprise to me looking at the PTP - again any difference in the assisted flow trigger or sedation?

Response: This sentence has been removed for clarity. As previously discussed, we did not observe difference in total RR in the BILEVEL-50-PSV50% group. However, we observed a different distribution of RR according to distinct phases of BIVENT. 

As previously mentioned, flow trigger sensitivity was adjusted at BASELINE-PEEP (INITIAL) for adequate inspiratory effort, according to esophageal pressure (Pes) decay. No additional changes to flow trigger sensitivity were done at any point in the experiment. The depth of the anesthesia was monitored via mean arterial pressure, heart rate and respiratory rate throughout the experiment.

---

## [Decision Letter · Decision Letter 1]

29 Jul 2021

Impact of different frequencies of controlled breath and pressure-support levels during biphasic positive airway pressure ventilation on the lung and diaphragm in experimental mild acute respiratory distress syndrome

PONE-D-20-40795R1

Dear Dr. Rocco,

We’re pleased to inform you that your manuscript has been judged scientifically suitable for publication and will be formally accepted for publication once it meets all outstanding technical requirements.

Kind regards,

Andrea Coppadoro

Academic Editor

PLOS ONE

Additional Editor Comments:

PLEASE CORRECT A TYPO IN THE ABSTRACT, FIRST LINE OF THE RESULTS, should read : bivent 50+psv0, compared to bivent 100 psv0, reduced...

Reviewers' comments:

Reviewer's Responses to Questions

**Comments to the Author**

1. If the authors have adequately addressed your comments raised in a previous round of review and you feel that this manuscript is now acceptable for publication, you may indicate that here to bypass the “Comments to the Author” section, enter your conflict of interest statement in the “Confidential to Editor” section, and submit your "Accept" recommendation.

Reviewer #2: All comments have been addressed

Reviewer #3: (No Response)

2. Is the manuscript technically sound, and do the data support the conclusions?

Reviewer #2: Yes

Reviewer #3: Yes

3. Has the statistical analysis been performed appropriately and rigorously? 

Reviewer #2: Yes

Reviewer #3: Yes

4. Have the authors made all data underlying the findings in their manuscript fully available?

Reviewer #2: Yes

Reviewer #3: Yes

5. Is the manuscript presented in an intelligible fashion and written in standard English?

Reviewer #2: Yes

Reviewer #3: Yes

6. Review Comments to the Author

Reviewer #2: I think that this study is very interesting. The authors answered completely all my questions. For these reason my suggestion is to accept this article.

Reviewer #3: I congratulate with the authors on the work provided during the revision process. I am satisfied with the responses provided by the authors. I just recommend to the authors to upload the images of the manuscript using a higher quality resolution as most of them are blurried.

7. PLOS authors have the option to publish the peer review history of their article (what does this mean?). If published, this will include your full peer review and any attached files.

Reviewer #2: No

Reviewer #3: No

---

## [Editor Report · Acceptance letter]

12 Aug 2021

PONE-D-20-40795R1 

Impact of different frequencies of controlled breath and pressure-support levels during biphasic positive airway pressure ventilation on the lung and diaphragm in experimental mild acute respiratory distress syndrome 

Dear Dr. Rocco:

I'm pleased to inform you that your manuscript has been deemed suitable for publication in PLOS ONE. Congratulations! Your manuscript is now with our production department. 

Kind regards, 

on behalf of

Dr. Andrea Coppadoro 

Academic Editor

PLOS ONE